# PhyMix: Towards Physically Consistent Single-Image 3D Indoor Scene Generation with Implicit–Explicit Optimization

## Abstract

Existing single-image 3D indoor scene generators often produce results that look visually plausible but fail to obey real-world physics, limiting their reliability in robotics, embodied AI, and design. To examine this gap, we introduce a unified Physics Evaluator that measures four main aspects: contact, stability, geometric priors, and deployability, which are further decomposed into nine sub-constraints, establishing the first benchmark to measure physical consistency. Based on this evaluator, our analysis shows that state-of-the-art methods remain largely physics-unaware. To overcome this limitation, we further propose a framework that integrates feedback from the Physics Evaluator into both training and inference, enhancing the physical plausibility of generated scenes. Specifically, we propose **PhyMix**, which is composed of two complementary components: (i) *implicit alignment* via Scene-GRPO, a critic-free group-relative policy optimization that leverages the Physics Evaluator as a preference signal and biases sampling towards physically feasible layouts, and (ii) *explicit refinement* via a plug-and-play Test-Time Optimizer (TTO) that uses differentiable evaluator signals to correct residual violations during generation. Overall, our method unifies evaluation, reward shaping, and inference-time correction, producing 3D indoor scenes that are both visually faithful and physically plausible. Extensive evaluations on synthetic dataset confirm state-of-the-art performance in both visual fidelity and physical plausibility, and extensive qualitative examples on stylized and real-world images further showcase the method's robustness.

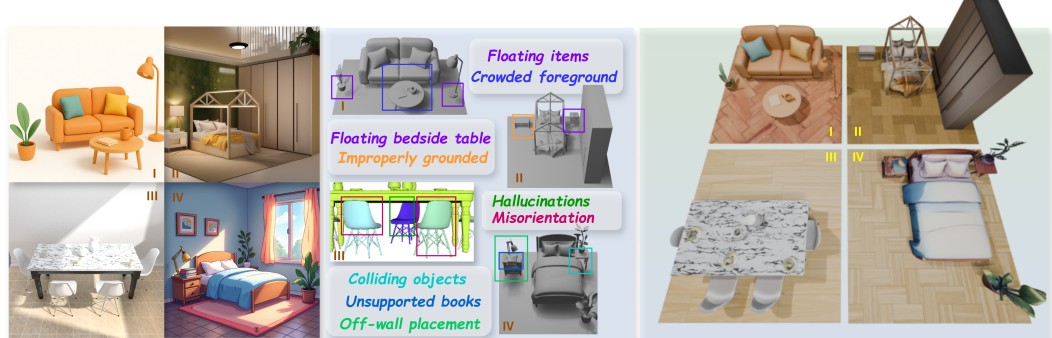

| Single-view Input | Existing Methods Performance | Ours (PhyMix) |
| --- | --- | --- |

Figure 1: **Comparison of single-image indoor scene generation.** Given a single-view input, including LLM-generated imagery, synthetic datasets, real-world photographs, or cartoon-style renderings(left), even the current advanced image-to-scene generation methods (Huang et al., 2025; Lin et al., 2025) frequently suffer from collisions, floating artifacts, misorientations, hallucinations, and unsupported objects. These issues often lead to unstable or cluttered arrangements, undermining physical plausibility. In contrast, our method, **PhyMix**, generates physically consistent and visually faithful 3D scenes with collision-free, grounded, and stable arrangements, while preserving high-fidelity object geometry.

## 1 INTRODUCTION

Generating 3D indoor scenes from a single image is a fundamental problem in computer vision and graphics, with applications spanning embodied AI, AR/VR design, and large-scale simulation (Dahnert et al., 2024; Liang et al., 2025). In these domains, accurate reconstructions are crucial for enabling physical interaction, supporting realistic digital content creation, and providing scalable environments for training and evaluation of robotic applications (Long et al., 2025; Wong et al., 2025). For applications involving physical interaction, scene reconstructions must extend beyond visual realism to guarantee physical plausibility (Xie et al., 2024). This means reconstructed scenes should not only appear realistic but also conform to physical constraints such as collision-free, support, and stability, which are important for downstream applications (Meng et al., 2025a).

There has been steady progress in single-image scene generation. One line of work (Ardelean et al., 2024; Han et al., 2024; Yao et al., 2025) uses a sequential pipeline, generates objects in isolation, then defines a layout to compose the scene, which, due to independently optimized stages, often yields inaccurate layouts. Another line of work (Chou et al., 2022; Ju et al., 2024; Zhou et al., 2024; Wang et al., 2022) predicts the entire scene as a unified voxel grid or implicit field, improving global coherence but sacrificing interactivity, since objects and background are fused into a single representation and cannot be manipulated independently. Lastly, instead of producing the objects and layouts separately, (Lin et al., 2025; Xiang et al., 2025; Huang et al., 2025; Meng et al., 2025b) attempts to predict both of them jointly. Nevertheless, as Figure 1 shows, they still struggle with physical plausibility, frequently yielding collisions, floating objects, and contact violations. As a result, ensuring physical plausibility in scene remains an open and challenging research problem.

While some recent methods include certain simple physical constraints, as summarized in Table 1, most existing methods handle physical consistency in an ad hoc manner, for example, simply penalizing collisions or checking whether objects touch the ground. What is still lacking is a comprehensive treatment of physical consistency that goes beyond isolated losses, together with a systematic empirical analysis of how such primitive signals affect plausibility. To address this gap, we propose a comprehensive Physics Evaluator that systematically covers four aspects: Contact, Stability, Geometric priors, and Deployability. These aspects are further decomposed into a total of nine measurable constraints. It provides a consistent way to assess physical plausibility, and its scores align closely with human judgments of physical plausibility. Because some of the constraints are differentiable, i.e., Contact and Stability while others are inherently non-differentiable, i.e., Geometric Priors and Deployability, integrating them into learning pipelines is nontrivial. Accordingly, we develop mechanisms to incorporate these signals in both training and inference stages.

Table 1: Comprehensive benchmark of physics-aware capabilities. We decompose physical consistency into four aspects: **Geometric Priors**, **Contact**, **Stability**, and **Deployability**. These aspects are further refined into nine measurable constraints. This unified protocol provides a comprehensive evaluation of physical plausibility and informs the design of our PhyMix approach.

| Method | Geometric Priors | | Contact | | | | Stability | | Deployability |
|---|---|---|---|---|---|---|---|---|---|
| | Orientation | Scale robustness | Collision free | Grounding | Support | Anchoring | Static stability | Dynamic stability | Reachability |
| CAST (Yao et al., 2025) | | | ✓ | ✓ | ✓ | | | | |
| DepR (Zhao et al., 2025) | | | | | | | | ✓ | |
| PhyScene (Yang et al., 2024) | | | ✓ | ✓ | | | | | ✓ |
| RoomCraft (Zhou et al., 2025a) | ✓ | | | ✓ | | | | | |
| HiScene (Dong et al., 2025) | ✓ | | | | | | | | |
| Gen3DSR (Ardelean et al., 2024) | | | | | | | | | |
| MIDI (Huang et al., 2025) | | | | | | | | | |
| PartCrafter (Lin et al., 2025) | | | | | ✓ | | | | |
| SceneGen (Meng et al., 2025b) | | ✓ | ✓ | | | | | | |
| CHOrD (Su et al., 2025) | ✓ | | ✓ | ✓ | | | | | |
| **PhyMix (Ours)** | ✓ | ✓ | ✓ | ✓ | ✓ | ✓ | ✓ | ✓ | ✓ |

Motivated by the above observations and our proposed Physics Evaluator, we present PhyMix (Physics-guided Implicit–Explicit Optimization). PhyMix builds on existing diffusion-based scene generation backbones and augments them with evaluator feedback in two complementary ways: Scene-GRPO (Scene-level Group Relative Policy Optimization) for implicit optimization during training, and TTO (Test-Time Optimizer) for explicit refinement during inference. Together, these components combine implicit and explicit strategies to enforce physical consistency. In particu-

lar, Scene-GRPO provides implicit physics optimization to address the challenge of incorporating non-differentiable physical constraints into diffusion training. Because non-differentiable physical constraints are hard to embed directly, we instead treat evaluator scores as preference signals, following preference alignment in large language models to inject our physical constraints, especially the non-differentiable ones, into the model. For each input, the model generates multiple candidate layouts that are ranked by the evaluator. The model is then optimized to assign higher probability to physically feasible layouts, gradually shifting its sampling distribution toward more physically plausible outcomes. Complementarily, the test-time optimizer performs explicit optimization, injecting differentiable physical constraints from the evaluator during denoising inference to adjust object poses and resolve potentially remaining violations. Together, these components unify implicit and explicit optimization under our evaluator, substantially improving physical plausibility while maintaining visual fidelity.

Our experiments show that PhyMix achieves superior physical consistency over prior methods, with fewer contact and stability failures, more coherent geometry, and improved navigability, while maintaining state-of-the-art visual fidelity. On the large-scale 3D-FRONT benchmark (Fu et al., 2021), our method raises the overall physics score by +20.2% relative to MIDI (Huang et al., 2025) and +16.6% relative to PartCrafter (Lin et al., 2025), delivering improved visual realism, highlighting the importance of embedding physical guidance into generation. These consistent gains across both physics and geometry confirm that embedding evaluator-driven implicit–explicit optimization yields state-of-the-art single-image 3D scene generation.

Our contributions can be summarized as:

- We introduce a comprehensive physics evaluator that decomposes physical consistency into four aspects and can be further categorized into nine measurable physical constraints.

- We propose PhyMix, which integrates implicit optimization (Scene-GRPO) to incorporate our physical constraints, especially the non-differentiable ones, and explicit optimization (TTO) to further refine differentiable ones, thereby yielding physically feasible generation.

- Our method outperforms existing approaches on the synthetic 3D-FRONT dataset, improving physical plausibility while preserving visual fidelity, and it generalizes to out-of-distribution inputs, including real-world photos, cartoon-style images, and LLM-generated paintings.

## 2 RELATED WORK

**3D indoor scene generation.** Existing approaches to single-image indoor scene generation can be roughly divided into three families: step-by-step assembly pipelines (Ardelean et al., 2024; Han et al., 2024; Yao et al., 2025; Yu et al., 2025), room-scale predictors (Liang et al., 2025; Chou et al., 2022; Ju et al., 2024; Zhou et al., 2024; Wang et al., 2022), and unified object–layout generators (Lin et al., 2025; Xiang et al., 2025; Huang et al., 2025; Meng et al., 2025b). As discussed in Section 1, each line of work has made steady progress but still struggles with physical plausibility. This motivates our focus on systematically incorporating physical consistency into scene generation.

**Physical plausibility in scene.** Early layout models such as ATISS (Paschalidou et al., 2021) and PlanIT (Wang et al., 2019) judged feasibility mainly by checking object overlaps. More recent methods introduced additional signals, such as collision-aware losses in diffusion sampling (Yang et al., 2024), constraint-based optimization (Su et al., 2025), or metrics for navigability and support (Tam et al., 2025). Others explored contact losses (Yao et al., 2025), relation constraints (Han et al., 2024), or reinforcement-style refinements (Liu et al., 2024). While these efforts show incremental gains, they remain ad hoc—focusing on isolated constraints without a unified treatment of physical consistency. This motivates the development of a systematic evaluator that can cover multiple aspects of physical plausibility in a systematic way.

**Preference-based alignment and test-time optimization.** In large language models, preference-based alignment (e.g., DPO (Rafailov et al., 2023), ORPO (Hong et al., 2024), GRPO (Shao et al., 2024)) has shown strong ability to guide generation without an explicit critic, and Diffusion-DPO extends this paradigm to image synthesis (Wallace et al., 2024). Conversely, PPO (Schulman et al., 2017) and AWR (Peng et al., 2019) assume sequential control with value

functions and stepwise rewards, which are absent in single-step 3D scene generation; applying them thus introduces unnecessary value estimation and ill-posed credit assignment. GRPO instead operates directly on ranked groups of final layouts, providing stable, low-variance updates without rollouts or critics. Unlike pairwise preference methods such as DPO, it also leverages multiple candidates per group to obtain richer preference signals. In parallel, test-time optimization methods refine layouts after generation. For example, flow-matching models (Lipman et al., 2022; 2024; Liu et al., 2022), constraint-based refinement in DiffuScene (Ju et al., 2024) and InstructScene (Lin & Mu, 2024), or reinforcement-style post-hoc adjustments in HAISOR (Sun et al., 2024). These approaches improve results but remain ad hoc and heuristic. Our method instead unifies the two directions: implicit GRPO alignment during training and explicit refinement at test time, both guided by our Physics Evaluator.

## 3 ARE EXISTING 3D INDOOR SCENE GENERATORS PHYSICALLY RELIABLE?

### 3.1 REPRESENTATIVE BASELINES ACROSS THE METHODOLOGICAL SPECTRUM

A central question of our study is: *Do existing single-image 3D indoor scene generators produce layouts that are physically reliable, or do they only look plausible to the eye?* To investigate this, we evaluate five representative baselines spanning the three major categories outlined in Section 1.

Step-by-step assembly methods such as **Gen3DSR** (Ardelean et al., 2024) and **REPARO** (Han et al., 2024) reconstruct objects individually and then place them into a scene. Room-scale approaches like **DepR** (Zhao et al., 2025) predict the entire layout at once. Unified object–layout generators, including **MIDI** (Huang et al., 2025) and **PartCrafter** (Lin et al., 2025), jointly model geometry and arrangement. Together, these baselines form a representative testbed for assessing physical reliability in single-image 3D scene generation.

### 3.2 MOTIVATION FOR A PHYSICS EVALUATOR

Despite impressive progress in visual plausibility and semantic coherence, existing scene generators still produce layouts with obvious physical errors: as shown in Figure 1, chairs sink into tables, lamps float above floors, and shelves collapse without support. Such violations not only break immersion but also make scenes unreliable for robotics, embodied AI, and design applications where physical interaction is critical. A key limitation of prior work is the absence of a standardized way to evaluate physical validity. While some prior works (Jin et al., 2024; Zhou et al., 2025b) employ differentiable rigid-body simulation frameworks to model gravity, inertia, and inter-object interactions, real physical simulation remains fundamentally limited to optimizing differentiable signals: it focuses on multi-step temporal dynamics, incurs heavy computational cost, and cannot capture many non-differentiable yet essential static plausibility factors such as orientation priors, class-level scale consistency, and reachability. To address this gap, we introduce a **Physics Evaluator** that systematically measures whether generated scenes obey physical constraints. It provides both comprehensive metrics for benchmarking and learning guidance for optimization, offering a unified tool to assess existing methods and to guide the generation of physically consistent scenes.

### 3.3 UNIFIED PHYSICS EVALUATOR

To systematically diagnose the failures observed across baselines, we introduce a **Physics Evaluator** that standardizes the assessment of physical consistency in single-image scene generation. These four aspects are organized in a coherent progression: starting from *geometric priors*, which ensure each object individually satisfies basic geometric constraints such as upright orientation, category-consistent size, and plausible aspect ratios; extending to *contact*, which evaluates inter-object and object–environment relationships including collisions, floor support, and alignment with room walls or canonical axes; further to *stability*, which tests whether objects remain well supported and maintain equilibrium when subjected to simple physics-based perturbations; and finally to *deployability*, which verifies that the resulting layout preserves sufficient free space and unobstructed regions for navigation and downstream embodied tasks. These four aspects, together with the clear and fine-

grained definitions of their internal sub-metrics detailed in Appendix A, form a unified benchmark for judging physical plausibility and provide structured signals that can guide both training and inference, as discussed in Section 4.

## 3.4 EMPIRICAL BENCHMARKING OF PHYSICAL CONSISTENCY

We apply the proposed Physics Evaluator to five representative baselines, establishing the first systematic benchmark of physical consistency for single-image scene generation (see Table 2 for details). To validate that these metrics capture physical plausibility, we conducted a perceptual study on static scene renderings with standardized viewpoints, randomized order, and method blinding, detailed in Appendix E. Raters provided 1–7 Mean Opinion Scores (MOS) and gave paired A/B preferences for the "more physically plausible" result. We report the macro-averaged win rate of *Proposed* vs. the strongest baseline, counting ties as 0.5. Our evaluator aligns well with human judgments, confirming that the metrics faithfully capture physical plausibility. *PhyMix* achieves the best performance across both metrics and user studies as described in Section 4.

Table 2: Benchmarking physical consistency across representative baselines using evaluator metrics and a perceptual user study. PhyMix ranks highest in both evaluations, with user preferences (Overall MOS) strongly aligned with metric scores (Overall Physical Score).

| Metric | MIDI | PartCrafter | Gen3DSR | DepR | REPARO | **Ours (PhyMix)** |
|---|---|---|---|---|---|---|
| *Evaluator: Contact* | | | | | | |
| Collision Rate (%) | 7.14 | 3.88 | 16.32 | 9.52 | 13.55 | **0.56** |
| Collision Severity (%) | 2.72 | 3.16 | 7.76 | 4.37 | 5.74 | **0.05** |
| Floating Rate (%)* | 33.15 | 28.39 | 45.39 | 37.06 | 42.46 | **0.97** |
| Floating Severity (%)* | 29.90 | 23.28 | 36.47 | 32.73 | 36.60 | **1.10** |
| Unanchored Rate (%) | 12.83 | 9.24 | 28.15 | 19.48 | 25.31 | **1.80** |
| *Evaluator: Stability* | | | | | | |
| Static Instability Rate (%) | 22.61 | 15.84 | 37.80 | 34.42 | 37.04 | **1.27** |
| Dynamic Instability Rate (%) | 24.63 | 18.92 | 41.27 | 36.95 | 39.72 | **2.20** |
| *Evaluator: Geometric Priors* | | | | | | |
| Misorientation Rate (%) | 7.37 | 5.82 | 14.29 | 11.83 | 13.95 | **1.72** |
| Scale Instability Rate (%) | 6.41 | 4.73 | 12.58 | 9.85 | 11.64 | **1.82** |
| *Evaluator: Deployability* | | | | | | |
| Unreachable Rate (%) | 3.78 | 2.94 | 8.67 | 6.51 | 7.89 | **1.01** |
| **Overall Physical Score** | 84.4 | 88.4 | 75.1 | 79.5 | 76.4 | **98.6** |
| *User Study* † | | | | | | |
| Contact MOS | 4.2 | 4.6 | 3.1 | 3.8 | 3.4 | **6.9** |
| Stability MOS | 4.1 | 4.8 | 2.9 | 3.5 | 3.2 | **6.8** |
| Geometric Prior MOS | 4.7 | 5.1 | 3.6 | 4.2 | 3.9 | **6.9** |
| **Overall MOS** | 4.3 | 4.8 | 3.2 | 3.8 | 3.5 | **6.8** |

| *Pairwise Preference (win rate)* | |
|---|---|
| PhyMix vs. PartCrafter | 0.92 (N=200, 95% CI [0.87, 0.95]) |
| PhyMix vs. MIDI | 0.96 (N=200, 95% CI [0.92, 0.98]) |

*"Floating" aggregates grounding and support violations. † Higher MOS indicates better perceptual quality. Preference values represent the scene-macro win rate, where ties are counted as 0.5.*

This study highlights a persistent gap: existing generators capture visual plausibility but struggle with reliable physical arrangement. Building on these findings, we incorporate evaluator feedback directly into training and inference to improve physical plausibility while preserving geometric and visual fidelity.

## 4 PHYMIX

Motivated by our benchmark findings, we introduce **PhyMix**, a framework that leverages feedback from our Physics Evaluator to enhance 3D scene physical consistency. Given a single RGB image $I$ and instance masks, our framework is based on a pretrained scene generator (e.g., PartCrafter (Lin et al., 2025) or MIDI (Huang et al., 2025)) predicts a scene

$$S = \{(z_i, \psi_i)\}_{i=1}^N, \tag{1}$$

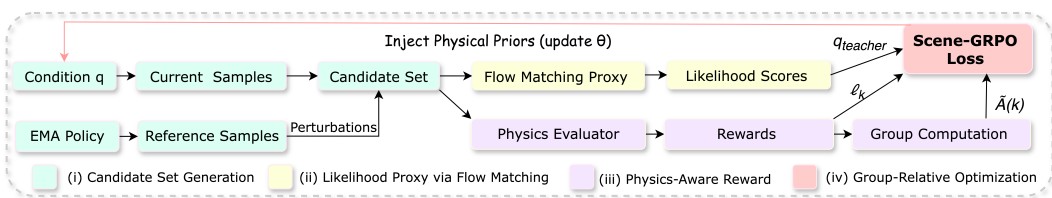

Figure 2: Overview of the architecture. The pipeline takes a single RGB image and instance masks as input and outputs physically consistent 3D scenes. Our PhyMix framework integrates (a) a unified Physics Evaluator that provides (b) implicit training-time rewards (Scene-GRPO) and explicit test-time guidance (TTO) to inject physical priors.

where each object is represented by a geometry latent $z_i$ and a pose $\psi_i = (t_i, q_i, s_i)$ in the world frame $W$. We then optimize the layout policy $\pi_\theta(S \mid I)$ with evaluator-driven rewards and penalties, without altering the underlying generator architecture, making PhyMix compatible with different backbones. Our overall objective couples a *training-time* expectation, combining evaluator rewards with a training-aligned regularizer, and an *inference-time* explicit physical penalty:

$$\max_\theta \ \mathbb{E}_{S \sim \pi_\theta(\cdot|I)} \Big[ \underbrace{r_{\text{geom}}(S) - \lambda\,\Phi(S)}_{R_{\text{physEval}}(S)} + \beta \, \log \pi_\theta(S \mid I) \ - \ \gamma \, \mathcal{E}_{\text{phys}}(S) \Big]. \tag{2}$$

Where $r_{\text{geom}}(S)$ encourages alignment with ground-truth layouts by rewarding accurate positions, orientations, and scales, while $\Phi(S)$ aggregates violation penalties defined by our Physics Evaluator across four categories: contact, stability, geometric plausibility, and deployability. Together these two terms form the evaluator score $R_{\text{physEval}}(S)$: the first promotes plausibility, and the second explicitly penalizes violations. The additional $\log \pi_\theta(S \mid I)$ term follows the flow-matching perspective (Lipman et al., 2022; 2024), acting as a likelihood proxy. It leverages the generative prior learned by the scene model through denoising-based training to ensure that predicted layouts remain consistent with the data distribution. Finally, $\mathcal{E}_{\text{phys}}(S)$ applies differentiable evaluation terms at inference for explicit correction. In summary, this unified formulation delivers complementary alignment: *implicit, distribution-level optimization* during training via critic-free GRPO, and *explicit, sample-level refinement* during inference through TTO.

## 4.1 SCENE-GRPO: IMPLICIT GROUP-RELATIVE POLICY 3D SCENE OPTIMIZATION

Figure 3: **Scene–GRPO** framework. (i) Candidate scenes are sampled from the current policy and from an Exponential Moving Average (EMA) (Tarvainen & Valpola, 2017) reference policy with added perturbations. (ii) Each candidate is assigned a likelihood score via a flow-matching proxy. (iii) Scenes are further evaluated with the Physics Evaluator to obtain physics-aware rewards, followed by group-wise comparison. (iv) Policy parameters are then updated with a critic-free GRPO objective, regularized by a KL term to ensure stability.

**Overview.** Scene-GRPO provides implicit alignment by biasing the policy distribution toward physically plausible layouts. As illustrated in Figure 3, for each condition we (i) form a candidate set by combining current policy samples with lightly perturbed reference scenes, (ii) approximate likelihood using a flow-matching proxy, (iii) evaluate physics-aware group-relative rewards, and (iv) optimize a critic-free group-relative objective, regularized by a KL term that keeps the policy close to the generative prior. This implicit strategy shifts the policy distribution toward scenes with higher physical scores, while maintaining training stability, complementing the explicit test-time refinement introduced later.

**Candidate Set Generation.** For each condition $q$, we construct a candidate set $\{S^{(k)}\}_{k=1}^{K}$ using two sources: (i) samples drawn from a pretrained scene generator $\pi_\theta(S \mid I)$, and (ii) slightly perturbed reference layouts. The reference layouts $S^\star$ are taken from an Exponential Moving Average (EMA) (Tarvainen & Valpola, 2017) version of the generator, which averages model parameters over time and therefore gives smoother and more stable outputs.

To introduce diversity, we apply small random shifts to object position, size, and rotation: translation is perturbed within a bounded range, scales are adjusted within class-dependent limits, and rotations are slightly varied around the upright axis. These perturbation ranges are gradually reduced during training, so that candidate sets evolve from diverse variations to mostly clean policy samples.

This design produces compact yet varied groups of scenes around stable references, allowing group-wise comparisons to be made directly from evaluator feedback, without needing an extra learned value function.

**Likelihood Proxy via Flow Matching.** To rank candidate scenes, we seek an estimate of their relative likelihood under the generator. Direct log-likelihood is hard to compute for diffusion-style models, so we follow the flow-matching view of generative modeling (Lipman et al., 2022), which shows that training with denoising objectives implicitly approximates maximum likelihood.

Specifically, we encode each scene $\{(z_i, \psi_i)\}_{i=1}^{N}$ into a latent $\mathbf{h}$, add Gaussian noise $\boldsymbol{\epsilon}$ at a random time step $t$, and train the transformer to predict this noise $\hat{\boldsymbol{\epsilon}}$:

$$\mathcal{L}_{\mathrm{FM}} = \|\hat{\boldsymbol{\epsilon}} - \boldsymbol{\epsilon}\|_2^2. \tag{3}$$

The negative loss $-\mathcal{L}_{\mathrm{FM}}$ serves as a differentiable proxy for $\log \pi_\theta(S \mid I)$, providing a practical score for comparing candidates within a group.

This proxy is motivated by the fact that better-aligned samples yield lower prediction error, hence higher implicit likelihood. In practice, it offers two advantages: (i) it is fully training-aligned, requiring only the same denoising step used in model learning, and (ii) it enables stable and consistent relative likelihood comparisons without expensive density estimation. Empirical validation (Appendix C) further confirms that this proxy correlates well with ground-truth likelihood estimates, making it effective for preference-based optimization.

**Physics-Aware Reward.** To balance geometric fidelity and physical plausibility, we define a reward for each candidate $S^{(k)}$ that has two components: (i) penalties for geometric misalignment, and (ii) penalties for physical violations measured by the evaluator:

$$r^{(k)} = -\left(w_c\,\mathrm{RMSE}_c + w_o\,\mathrm{MAE}_\theta/180° + w_s\,\mathrm{RelErr}_s\right) - \sum_j \lambda_j\,\Delta P_j(S^{(k)}), \tag{4}$$

where the first term penalizes translation, rotation, and scale errors, and the second term aggregates nine violation terms from the Physics Evaluator, covering contact (collision, grounding, support, anchoring), stability (static and dynamic), geometric plausibility (orientation and scale), and deployability (reachability). For each violation term $\Delta P_j$, we measure the relative increase of the corresponding evaluator penalty compared to a reference scene, with small tolerances to avoid over-penalization. Detailed definitions of all factors $P_j$ are provided in Appendix A.

**Group-Relative Objective and Optimization.** To align the policy with physical feasibility, we adopt a group-relative, critic-free objective inspired by recent preference-optimization methods (Shao et al., 2024). The key idea is to form candidate sets that mix standard policy samples with perturbed references, and then update the model based on their relative ranking under physics-aware rewards and likelihood proxies. Let $\ell^{(k)}$ be the flow-matching loss for candidate $S^{(k)}$, and let $\tilde{A}^{(k)} = \frac{r^{(k)} - \bar{r}}{\mathrm{std}(r)}$ denote the groupwise $z$-score advantage. We then minimize:

$$\mathcal{L}_{\mathrm{GRPO}} = \frac{1}{K} \sum_{k=1}^{K} \ell^{(k)}\left(-\tilde{A}^{(k)}\right) + \beta \cdot \mathrm{KL}(p_\theta \| q), \qquad p_\theta = \mathrm{softmax}\left(\frac{-\ell}{\tau}\right), \;\; q = \mathrm{softmax}\left(\frac{r}{\tau}\right), \tag{5}$$

which reallocates probability mass toward higher-reward layouts without requiring a value network. The additional KL term distills the policy distribution $p_\theta$ toward a reward-induced teacher $q$, improving stability and sample efficiency. In practice, we use groups of size $K = 12$, apply a target-KL scheduler for $\beta$, and maintain an EMA reference policy with momentum 0.99.

Thus, this Scene-GRPO implicitly biases the sampling distribution toward physically feasible scenes through groupwise comparisons. Importantly, this preference-based formulation does not require differentiability of the evaluator: even non-differentiable physical metrics can be incorporated as relative rewards, enabling effective optimization in scenarios where gradient-based losses fail.

### 4.2 EXPLICIT TEST-TIME OPTIMIZATION

While Scene-GRPO guides the model toward physically plausible layouts, residual violations can still appear at inference, mainly due to monocular scale ambiguity and complex object interactions. To handle these cases, we introduce a lightweight test-time optimization module that explicitly enforces physics during denoising. It focuses on two aspects, *contact* and *stability*, because these constraints are differentiable and thus provide reliable gradient signals. Other factors (e.g., orientation, class-level scale, reachability) are addressed implicitly by Scene-GRPO.

Concretely, at selected timesteps we decode each object into a coarse signed distance field (SDF) proxy and minimize a differentiable energy:

$$\mathcal{E}_{\text{phys}} = \alpha_{\text{col}}\mathcal{E}_{\text{col}} + \alpha_{\text{grd}}\mathcal{E}_{\text{grd}} + \alpha_{\text{anc}}\mathcal{E}_{\text{anc}} + \alpha_{\text{stb}}\mathcal{E}_{\text{stb}} + \alpha_{\text{reg}}\mathcal{E}_{\text{reg}}. \tag{6}$$

Where $\mathcal{E}_{\text{col}}$ penalizes overlaps, $\mathcal{E}_{\text{grd}}$ attracts objects toward the ground, $\mathcal{E}_{\text{anc}}$ enforces wall alignment, $\mathcal{E}_{\text{stb}}$ promotes static stability, and $\mathcal{E}_{\text{reg}}$ smooths gradients, detailed in Appendix D. All thresholds follow the Physics Evaluator for consistency between training rewards and test-time corrections. To stay efficient, optimization is applied only at a few key denoising phases, where a small number of gradient steps are interleaved with updates to correct physical inconsistencies.

This explicit module is plug-and-play, adds negligible overhead, and consistently improves physical plausibility, complementing the implicit alignment of Scene-GRPO.

## 5 EXPERIMENTS

### 5.1 EXPERIMENTAL SETTINGS

We conduct main experiments on 3D-FRONT (Fu et al., 2021) and assess cross-domain generalization on held-out modalities. *PhyMix* is implemented on multi-instance single-image 3D backbones with LoRA-based (Hu et al., 2022) fine-tuning. We evaluate physical consistency with our Physics Evaluator (Section 3.3) and geometry/visual fidelity using standard scene- and object-level metrics and compare against representative single-image scene-generation baselines across four method families. The details are in Appendix F.

### 5.2 COMPARISON WITH EXISTING WORKS

**Quantitative Comparison.** Table 3 reports results on the 3D-FRONT test set, covering both physics and geometry/visual metrics. We evaluate three variants of our method: training-time implicit alignment with Scene-GRPO only, explicit test-time optimization (TTO) only, and the full framework combining both. Scene-GRPO alone already improves performance across all metrics by biasing the distribution toward physically feasible layouts. TTO alone mainly benefits differentiable terms such as collision, grounding, and stability, but has a limited impact on non-differentiable terms such as misorientation, scale, and reachability. When combined, the two components complement each other: collision and floating rates drop below 1%, the overall physical score reaches 98.6, and geometric fidelity also improves with the lowest Chamfer Distance, highest F-Scores, and best IoU-B. These results highlight the complementary roles of GRPO and TTO: the former provides distribution-level alignment, while the latter refines individual samples at inference. Across all three variants, our approach consistently outperforms all the existing baselines.

**Qualitative Comparison.** Figure 4 shows results of our method *PhyMix* compared with representative baselines (MIDI, PartCrafter, Gen3DSR, DepR) on four styles of inputs. Baselines often exhibit diverse physical artifacts, including floating furniture, inter-object collisions, unstable or unsupported placements, and hallucinated or misoriented geometry, highlighted in colored boxes. In contrast, our method produces scenes that remain physically consistent across all input styles, with objects grounded, stably supported, and free of major collisions, while also preserving geometry and visual fidelity. This demonstrates that *PhyMix* generalizes beyond training distributions and delivers reliable physics-aware generation in both synthetic and real-world settings. More comparative experiments are provided in Appendix G.

Table 3: Comprehensive evaluation on the 3D-FRONT test set.

(a) Physics-based metrics

| Method | Collision↓ | Floating↓ | Unanchored↓ | Static Inst.↓ | Dynamic Inst.↓ | Misori.↓ | Scale Inst.↓ | Unreach.↓ | Overall↑ |
|---|---|---|---|---|---|---|---|---|---|
| PartCrafter (Lin et al., 2025) | 3.88 | 28.4 | 9.24 | 15.8 | 18.9 | 5.82 | 4.73 | 2.94 | 88.4 |
| DepR (Zhao et al., 2025) | 9.52 | 37.1 | 19.5 | 34.4 | 37.0 | 11.8 | 9.85 | 6.51 | 79.5 |
| Gen3DSR (Ardelean et al., 2024) | 16.3 | 45.4 | 28.2 | 37.8 | 41.3 | 14.3 | 12.6 | 8.67 | 75.1 |
| REPARO (Han et al., 2024) | 13.6 | 42.5 | 25.3 | 37.0 | 39.7 | 14.0 | 11.6 | 7.89 | 76.4 |
| MIDI (Huang et al., 2025) | 7.14 | 33.2 | 12.8 | 22.6 | 24.6 | 7.37 | 6.41 | 3.78 | 84.4 |
| Ours (+GRPO) | 2.32 | 6.73 | 3.85 | 3.33 | 4.15 | 2.18 | 1.95 | 1.42 | 96.7 |
| Ours (+TTO) | 4.85 | 8.10 | 6.10 | 7.95 | 8.70 | 7.35 | 6.10 | 3.55 | 93.2 |
| Ours (+GRPO+TTO) | **0.56** | **0.97** | **1.80** | **1.27** | **2.20** | **1.72** | **1.82** | **1.01** | **98.6** |

(b) Geometry/visual metrics

| Method | CD-S↓ | CD-O↓ | F-Score-S↑ | F-Score-O↑ | IoU-B↑ |
|---|---|---|---|---|---|
| PartCrafter (Lin et al., 2025) | 0.098 | – | 55.9 | – | – |
| DepR (Zhao et al., 2025) | 0.104 | 0.118 | 52.0 | 48.4 | 0.57 |
| Gen3DSR (Ardelean et al., 2024) | 0.103 | 0.126 | 41.7 | 40.4 | 0.41 |
| REPARO (Han et al., 2024) | 0.109 | 0.134 | 43.5 | 44.4 | 0.40 |
| MIDI (Huang et al., 2025) | 0.072 | 0.094 | 57.2 | 60.0 | 0.60 |
| Ours (+GRPO) | 0.044 | 0.055 | 70.5 | 67.5 | 0.69 |
| Ours (+TTO) | 0.049 | 0.053 | 68.5 | 64.2 | 0.66 |
| Ours (+GRPO+TTO) | **0.038** | **0.043** | **75.5** | **73.2** | **0.72** |

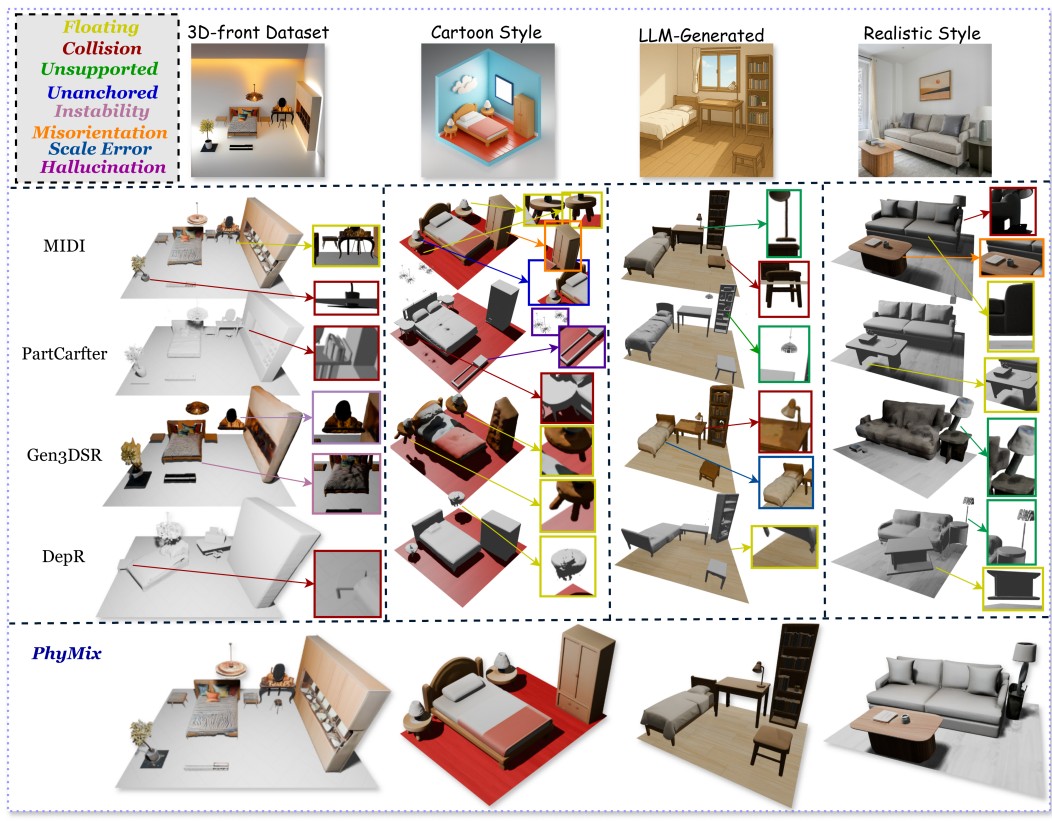

Figure 4: Qualitative comparison on 3D-FRONT, cartoon, LLM-generated, and real images. Colored boxes highlight different physical errors (see legend, top-left) in baselines. Our method *PhyMix* produces grounded, stable, and physically consistent layouts with preserved visual fidelity.

## 5.3 ABLATION STUDIES

The ablation study of GRPO and TTO are presented in Table 3: Ours (+GRPO), Ours (+TTO), and Ours (+GRPO+TTO). Here, we discuss additional ablation aspects, including (i) compatibility with

different 3D scene generative architectures, (ii) physics loss components in test-time optimization, and (iii) the effect of group size in Scene-GRPO.

Table 4: Ablation studies. For the loss ablation, we report contact indicators (Collision, Floating, Unanchored), and Static Instability. The group size study shows the trade-off among physical consistency and runtime.

(a) Cross-backbone generalization

| Backbone | Baseline Overall↑ | Ours Overall↑ | Δ |
|---|---|---|---|
| MIDI | 82.0 | **98.6** | +16.9 (20.2%) |
| PartCrafter | 85.9 | **96.3** | +10.4 (16.6 %) |

(b) Loss ablation

| Config | Collision↓ | Float↓ | Unanchored↓ | Stat.Inst.↓ |
|---|---|---|---|---|
| Full | **0.56** | **0.97** | **1.80** | **1.27** |
| w/o overlap | 2.24 | 0.98 | 1.92 | 1.48 |
| w/o ground-zone | 0.62 | 6.18 | 2.07 | 1.39 |
| w/o anchoring | 0.58 | 1.02 | 3.54 | 1.31 |
| w/o stability | 0.57 | 0.99 | 1.87 | 3.11 |

(c) Effect of group size $K$

| $K$ | Overall↑ | Runtime |
|---|---|---|
| 2 | 92.3 | 1.0× |
| 4 | 95.5 | 1.8× |
| 8 | 97.0 | 3.2× |
| 12 | **98.6** | 4.5× |
| 16 | 98.5 | 6.1× |

**Compatibility with 3D Scene Generative Architectures**    To verify that PhyMix is compatible to various 3D scene generative architectures, we apply Scene-GRPO to two representative and most-recent backbones: PartCrafter and MIDI. As shown in Table 4a, our method consistently improves physical plausibility: +16.9 points on MIDI (+20.2% relative) and +10.4 on PartCrafter (+16.6% relative), with both exceeding an overall score of 94. This shows that our approach is not tied to a specific architecture and can be applied to representative indoor scene generation models.

**Physics loss components.**    Table 4b reports results when removing individual energy terms from our physics evaluator during TTO. Removing the soft overlap energy substantially increases collisions, the ground-zone term is critical for avoiding floating objects, and the stability term has the strongest impact on balance. Together, these results confirm that all three components contribute complementary benefits.

**Group size in Scene-GRPO.**    We vary the candidate group size $K$ to examine optimization stability and quality. As shown in Table 4c, performance saturates around $K=12$, striking a balance between effectiveness and efficiency.

### 5.4    ADDITIONAL EXPERIMENTAL ANALYSES

To further demonstrate the robustness and generality of our method, we conduct additional experiments and report the detailed results in Appendix G. These include (i) extended visual comparisons with existing models across diverse input modalities; (ii) a validation of our Physics Evaluator against a Taichi-based rigid-body simulator, showing strong consistency across collision, grounding, and stability metrics; (iii) a detailed computational analysis comparing our inference cost with baseline models and with physics simulation; (iv) qualitative failure cases highlighting limitations with deformable objects and complex geometries in realistic image inputs; and (v) downstream navigation task demonstrations validating the practical utility of our reachability metric.

## 6    CONCLUSION

We presented PhyMix, a framework for physically consistent single-image 3D indoor scene generation. The proposed Physics Evaluator decomposes plausibility into four aspects and nine measurable metrics, serving both as a systematic benchmark and as guidance for optimization. Building on this, PhyMix integrates Scene-GRPO for implicit distribution-level alignment and test-time optimization (TTO) for explicit sample-level refinement, allowing evaluator feedback to be embedded into both training and inference. Experiments show that this combination delivers state-of-the-art physical plausibility and geometric fidelity, with results that align well with human perceptual judgments. Limitations are discussed in Appendix H.

## ETHICS STATEMENT

This work targets single-image 3D indoor scene generation with a physics-guided pipeline (*PhyMix*). We use publicly available indoor-scene datasets (e.g., 3D-FRONT (Fu et al., 2021)) and do not collect new human-subject data; no personally identifying content is required. Potential risks include privacy leakage when reconstructing private spaces and misuse for unauthorized mapping.

## REPRODUCIBILITY STATEMENT

We make reproduction feasible by fully specifying algorithms, metrics, and settings in the paper and appendices. In particular: (i) the unified Physics Evaluator and its thresholding rules are detailed in Appendix A, with differentiable physical penalties and gradient-stability strategies in Appendix D; (ii) implementation details, optimizer/schedules, Scene-GRPO group size $K$, and TTO schedules are enumerated in Appendix F.2; (iii) datasets and splits are documented in Appendix F; and (iv) all experiments use fixed random seeds, with hardware and compute budgets (GPU types, batch sizes, wall-clock estimates) reported in Appendix F.2. These materials are intended to enable faithful reproduction using publicly available resources. Codes will be released upon Publication.

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

# A  UNIFIED PHYSICS EVALUATOR

Our *Physics Evaluator* not only measures physical plausibility but also provides signals that improve scene quality in our framework. Earlier methods often relied on fixed thresholds to detect collisions, which offered limited guidance and were sensitive to scale. In contrast, our evaluator works in a consistent world frame with adaptive thresholds (Appendix B), making it robust to scale ambiguity in single-image inputs. It serves three purposes: (1) discrete mesh-based evaluation for benchmarking, (2) non-differentiable reward signals that guide Scene-GRPO training, and (3) differentiable surrogate terms that enable test-time refinement.

**Setup.** To make all physical checks consistent across scenes of different scales, we define an adaptive threshold that grows with room and object size. Specifically, we estimate the floor plane in the world frame and set a baseline threshold

$$\tau_{\text{adaptive}} = \max\big(\tau_{\text{ratio}} \cdot \text{diag}(\text{BB}_{\text{room}}),\ \tau_{\text{edge}} \cdot \overline{L}_{\text{edge}}\big), \tag{7}$$

where $\text{BB}_{\text{room}}$ is the room bounding box, $\text{diag}(\cdot)$ its diagonal length, and $\overline{L}_{\text{edge}}$ the average furniture edge length. The constants $\tau_{\text{ratio}}$ and $\tau_{\text{edge}}$ are dimensionless factors that convert these size measures into task-specific thresholds, ensuring that collision, contact, and stability checks are scale-aware rather than fixed.

**Physics Metrics.** We group our evaluation into four aspects: contact, stability, geometric priors, and deployability, listed together below.

(1) CONTACT METRICS. To capture whether objects are plausibly placed with respect to their surroundings, we define four contact-related indicators: collision, grounding, support, and anchoring. Formally,

$$P_{\text{coll}}(S) = \tfrac{1}{|F|} \sum_{j \in F} \mathbf{1}\big[\exists\, i \neq j : \text{Collision}(M_i, M_j)\big], \tag{8}$$

$$P_{\text{ground}}(S) = \tfrac{1}{|F|} \sum_{j \in F} \mathbf{1}\big[\min_{v \in V_j} |v_z - z_{\text{floor}}| < \tau_g\big], \tag{9}$$

$$P_{\text{sup}}(S) = \tfrac{1}{|F|} \sum_{j \in F} \mathbf{1}\big[\min_{v \in V_j} |v_z - z_{\text{support}(j)}| < \tau_s\big], \tag{10}$$

$$P_{\text{anc}}(S) = \tfrac{1}{|F|} \sum_{j \in F} \Big( \mathbf{1}\big[\angle(\text{axis}(M_j), \mathcal{A}_{\text{room}}) < \tau_c\big] + \mathbf{1}\big[d(M_j, \partial\mathcal{B}_{\text{room}}) < \tau_w\big]\Big)/2. \tag{11}$$

where $S$ is the generated scene; $F \subset S$ the set of furniture objects (excluding floor/walls); $M_j$ the $j$-th object with vertex set $V_j$; $z_{\text{floor}}$ the floor height; $z_{\text{support}(j)}$ the closest supporting plane height for $M_j$; $\text{axis}(M_j)$ the object's upright axis; $\mathcal{A}_{\text{room}}$ the Manhattan axes; $\partial\mathcal{B}_{\text{room}}$ the room boundary; $d(\cdot, \cdot)$ the shortest distance; $\angle(\cdot, \cdot)$ the angle operator; $\mathbf{1}[\cdot]$ the indicator; and $\tau_g, \tau_s, \tau_c, \tau_w$ are task thresholds linearly derived from $\tau_{\text{adaptive}}$.

(2) STABILITY METRICS. We measure static stability and short-horizon dynamical stability:

$$P_{\text{stab}}(S) = \tfrac{1}{|F|} \sum_{j \in F} \frac{d\big(\mathbf{c}_j^{\text{proj}},\ \text{ConvexHull}(P_{\text{contact}}^j)\big)}{\sqrt{\text{Area}(\text{ConvexHull}(P_{\text{contact}}^j))}}, \tag{12}$$

$$P_{\text{sim}}(S) = \tfrac{1}{|F|} \sum_{j \in F} \mathbf{1}\big[\text{PhysicsSim}(M_j, T) \text{ is stable}\big]. \tag{13}$$

where $\mathbf{c}_j^{\text{proj}}$ is the projection of the center of mass of $M_j$ onto the contact plane along gravity; $P_{\text{contact}}^j$ the set of contact points of $M_j$; $\text{ConvexHull}(\cdot)$ the convex-hull operator; $\text{Area}(\cdot)$ the hull area; $d(\cdot, \cdot)$ the signed distance to the hull boundary; $\text{PhysicsSim}(M_j, T)$ a rigid-body rollout over horizon $T$; "is stable" indicates no tipping, falling, or persistent sliding.

(3) GEOMETRIC PRIORS. We encourage consistency with gravity and class-level scale statistics:

$$P_{\text{mis}}(S) = \tfrac{1}{|F|} \sum_{j \in F} \mathbf{1}\big[\angle(\mathbf{n}_j, \mathbf{g}) > \tau_m\big], \tag{14}$$

$$P_{\text{scale}}(S) = \tfrac{1}{|F|} \sum_{j \in F} \mathbf{1}\Big[\big|\tfrac{\text{diag}(M_j)}{\mu_{c(j)}} - 1\big| < \tau_{sc}\Big]. \tag{15}$$

where $\mathbf{n}_j$ is the estimated upright axis of $M_j$; $\mathbf{g}$ the unit gravity direction; $\text{diag}(M_j)$ the diagonal length of the axis-aligned bounding box of $M_j$; $c(j)$ the semantic class of $M_j$; $\mu_{c(j)}$ the class-wise diagonal statistic; and $\tau_m, \tau_{sc}$ are the respective thresholds.

(4) DEPLOYABILITY. We evaluate deployability via navigation reachability. Specifically, we test whether a valid path exists between sampled start and goal positions on the floor plane using an $A^*$ planner:

$$P_{\text{reach}}(S) = \mathbf{1}\big[\text{A}^*(z_{\text{start}}, z_{\text{goal}}; \mathcal{O}) \neq \varnothing\big], \tag{16}$$

where $z_{\text{start}}, z_{\text{goal}}$ are grid points, $\mathcal{O}$ is the occupancy map derived from scene geometry, and $\text{A}^*(\cdot)$ returns a path if one exists. Scenes that block all feasible routes are penalized as unreachable.

**Adaptive thresholding.** We adopt similarity-invariant thresholding to eliminate the effect of global scaling:

$$\tau_{\text{adaptive}} = \max\big(\tau_{\text{ratio}} \cdot \text{diag}(\text{BB}_{\text{room}}), \ \tau_{\text{edge}} \cdot \overline{L}_{\text{edge}}\big). \tag{17}$$

where $\tau_{\text{ratio}} \in [0.002, 0.005]$ and $\tau_{\text{edge}} \in [0.3, 0.8]$ are calibrated ranges; in practice $\tau_{\text{ratio}} = 0.003$, $\tau_{\text{edge}} = 0.5$ work robustly. Task thresholds such as $\tau_g, \tau_s, \tau_c, \tau_w, \tau_m, \tau_{sc}$ are linearly derived from $\tau_{\text{adaptive}}$ so that binary decisions are invariant under global similarity transforms.

**Threshold sensitivity and calibration.** The adaptive threshold parameters are set as $\tau_{\text{ratio}} \in [0.002, 0.005]$ (0.2%-0.5% of room diagonal) and $\tau_{\text{edge}} \in [0.3, 0.8]$ (30%-80% of average edge length $\overline{L}_{\text{edge}}$). For floating detection, $\tau_f = \tau_{\text{adaptive}}$ prevents false positives from furniture legs while detecting genuine floating objects. Empirically, $\tau_{\text{ratio}} = 0.003$ and $\tau_{\text{edge}} = 0.5$ provide optimal balance across diverse scene scales, with sensitivity analysis showing $\pm 15\%$ variation in floating detection rates when parameters vary by $\pm 50\%$ from these values.

**Physics-informed rewards for Scene-GRPO.** We integrate physical consistency into preference-driven policy optimization with a composite reward balancing data fidelity and physical feasibility. For a predicted layout $\hat{S}$ and a reference $S^*$:

$$R(\hat{S}, S^*) = R_{\text{geom}}(\hat{S}, S^*) + R_{\text{phys}}(\hat{S}, S^*), \tag{18}$$

with the data term

$$R_{\text{geom}} = -\lambda_{\text{pos}} \cdot \text{Centroid\_RMSE} - \lambda_{\text{orient}} \cdot \tfrac{\text{Orient\_MAE}}{180°} - \lambda_{\text{scale}} \cdot \text{Scale\_RelErr}. \tag{19}$$

The physics term aggregates relative improvements over the reference across indicators:

$$\begin{aligned} R_{\text{phys}} = &-\lambda_{\text{coll}} \cdot \Delta P_{\text{coll}} - \lambda_{\text{ground}} \cdot \Delta P_{\text{ground}} - \lambda_{\text{sup}} \cdot \Delta P_{\text{sup}} - \lambda_{\text{anc}} \cdot \Delta P_{\text{boundary}} \\ &- \lambda_{\text{stab}} \cdot \Delta P_{\text{stab}} - \lambda_{\text{sim}} \cdot \Delta P_{\text{sim}} - \lambda_{\text{mis}} \cdot \Delta P_{\text{mis}} - \lambda_{\text{scale}} \cdot \Delta P_{\text{scale}} \\ &- \lambda_{\text{nav}} \cdot \Delta P_{\text{nav}} - \lambda_{\text{cons}} \cdot \Delta P_{\text{consistency}}. \end{aligned} \tag{20}$$

where $\Delta M = \max\big(0, \ M(\hat{S}) - M(S^*) - \delta\big)$ is the improvement beyond a slack $\delta$; $\lambda_{\bullet}$ are non-negative weights; $\text{Centroid\_RMSE}$, $\text{Orient\_MAE}$, and $\text{Scale\_RelErr}$ measure position, orientation, and scale discrepancies. To stabilize preference-based advantages, we normalize rewards across a group:

$$\tilde{R}_i = \frac{R_i - \mu_R}{\sigma_R}, \qquad \tilde{R}_i \in \mathbb{R}, \tag{21}$$

where $\mu_R, \sigma_R$ are the group mean and standard deviation.

Empirically, we set the geometric alignment weights as $(\lambda_{\text{pos}}, \lambda_{\text{orient}}, \lambda_{\text{scale}}) = (1.0, 0.25, 0.25)$; for the physics-related terms we use the same $\lambda$ for all components.

**Differentiable physical penalties for TTO.** During denoising, we decode coarse SDF grids at resolution $R \in \{12, 16, 20\}$ and compute differentiable penalties:

$$\mathcal{L}_{\text{col}} = \frac{1}{|\mathcal{V}|} \sum_{i<j} \left[ \alpha_1 \left\langle \sigma(-\phi_i/\tau), \sigma(-\phi_j/\tau) \right\rangle + \alpha_2 \left\| \text{ReLU}(-\phi_i) \odot \text{ReLU}(-\phi_j) \right\|_1 \right], \quad (22)$$

$$\mathcal{L}_{\text{grd}} = \sum_i \left[ \text{WeightedHeight}(\phi_i) + \text{GroundDeficit}(\phi_i) + \text{GroundVoid}(\phi_i) \right], \quad (23)$$

$$\mathcal{L}_{\text{anc}} = \sum_i \left[ \text{AxisAlign}(\phi_i; \mathcal{A}) + \text{WallContact}(\phi_i; \phi^{\text{wall}}) + \text{WallNoPenetration}(\phi_i; \phi^{\text{wall}}) \right], \quad (24)$$

$$\mathcal{L}_{\text{stb}} = \sum_i \text{CenterOfMassPenalty}(\phi_i), \qquad \mathcal{L}_{\text{reg}} = \sum_i \text{SDFSmoothness}(\phi_i). \quad (25)$$

where $\phi_i$ is the signed distance field of object $i$ in the decoded grid; $\sigma(\cdot)$ is the logistic function; $\tau$ controls the soft bandwidth; $\alpha_1, \alpha_2 > 0$ weight overlap probability and penetration energy; $P_{\text{contact}}^i$ is the contact set. To avoid gradient saturation, we truncate for $|\phi| < \epsilon$ and clip gradients during backpropagation through the SDF decoder. Complete formulations and stability strategies are detailed in Appendix D.

**Similarity-invariant thresholding.** For binary decisions such as floating or collision detection, invariance under global similarity transforms is necessary: if a scene is scaled by $s > 0$, the decision must be unchanged. Let $d$ be a world-space distance relevant to the test and let $\tau$ be the decision threshold. Under $x \mapsto sx$, $d \mapsto sd$. The decision $\mathbf{1}[d > \tau]$ is invariant iff $\tau$ scales linearly, i.e., $\tau' = s\tau$. Hence, any admissible rule must set $\tau$ proportional to a scene-dependent length scale. Our hybrid design $\tau_{\text{adaptive}} = \max(\tau_{\text{ratio}} \cdot \text{diag}(\text{BB}_{\text{room}}), \tau_{\text{edge}} \cdot \overline{L}_{\text{edge}})$ is the minimal linear construction that remains stable across both room-scale and object-scale variations.

## B  WORLD FRAME ANCHORING AND SCALE RECOVERY

Given a single RGB image $I$, we estimate an absolute metric scale, align the scene to a gravity-aligned world frame, and propagate the resulting scale uncertainty to downstream thresholds. We intentionally avoid heavy notation; only essential equations are kept.

**Scale from simple priors.** For each detected furniture instance, we divide a class-level expected physical height by its image bbox height to obtain a per-instance scale. We take the *median* over valid instances as the furniture scale $s_{\text{furn}}$. When the distances to ceiling and floor can be estimated in image units, we treat the nominal room height $H_{\text{target}}$ as a ruler and obtain a second scale $s_{\text{ceil}}$ by dividing $H_{\text{target}}$ by the ceiling–floor pixel distance. Uncertainty in $H_{\text{target}}$ maps directly to $\sigma_{\text{ceil}}$ through this ratio.

**Fusion and fallback.** If both priors are available, we combine them by *precision weighting* (weights proportional to $1/\sigma^2$), yielding $s_{\text{abs}}$ and $\sigma_{\text{abs}}$. If only furniture evidence is available, we apply a light *clipped multiplicative* fallback: we center at $s_{\text{furn}}$ with a bounded factor $c_{\text{fb}} \in [0.7, 1.4]$ (mean 1), and set a default relative uncertainty $\sigma_{\text{abs}} = \gamma s_{\text{abs}}$ with $\gamma = 0.15$.

**Gravity-aligned world frame.** We estimate the ground normal $\mathbf{n}_{\text{ground}}$ and rotate it onto the world up-axis $\mathbf{e}_z = [0, 0, 1]^\top$ using Rodrigues' formula to obtain $\mathbf{R}_{\text{align}}$. The Sim(3) transform to world coordinates is

$$\mathbf{v}_{\text{world}} = s_{\text{abs}} \mathbf{R}_{\text{align}} \left( \mathbf{v}_{\text{model}} - \mathbf{t}_{\text{model}} \right) + \mathbf{t}_{\text{world}}, \quad (26)$$

and we place the ground plane at $z = 0$.

**Uncertainty-aware thresholds.** For any length-like base threshold $\tau_{\text{base}}$, we inflate it by the relative scale uncertainty to remain conservative under multiplicative scale noise. Let $\text{cv} = \sigma_{\text{scale}}/s_{\text{abs}}$ (choosing $\sigma_{\text{scale}}$ from $\{\sigma_{\text{abs}}, \sigma_{\text{ceil}}, \sigma_{\text{furn}}\}$ according to availability). We set

$$\tau_{\text{robust}} = \tau_{\text{base}}\sqrt{1 + \text{cv}^2}, \tag{27}$$

which follows first-order uncertainty propagation and keeps thresholds invariant in expectation under global similarity transforms.

*Defaults.* $H_{\text{target}}$ is a nominal room-height prior. Robust statistics use all furniture detections that pass confidence and size sanity checks. The fallback parameters $(c_{\text{fb}}, \gamma)$ are fixed unless stated otherwise.

## C  REFERENCE EVIDENCE ESTIMATION AND PROXY VALIDATION

This appendix formalizes the construction of a *reference evidence* for candidate scenes and validates the *flow-matching* proxy used in the main text. Our goal is not to recover a calibrated ground-truth log-likelihood (intractable for our pipeline), but to demonstrate that the proxy preserves the *within-group ordering* required by the group-relative optimization in Section 4.1.

### C.1  PROBLEM SETTING AND NOTATION

Given a conditioning signal, the policy proposes a group of $K$ scene hypotheses $\{S^{(k)}\}_{k=1}^{K}$. The training objective in Section 4.1 depends on two candidate-wise quantities: a likelihood-related score derived from the flow-matching loss $\mathcal{L}_{\text{FM}}$ in Equation 3, and a physics-based reward $r^{(k)}$ in Equation 4. The GRPO loss in Equation 5 uses their *order* through groupwise advantages and a low-temperature teacher distribution. We therefore require: (i) a principled reference against which to assess likelihood proxies and (ii) a rigorous evaluation of rank agreement at the group level.

### C.2  IMPORTANCE-WEIGHTED REFERENCE EVIDENCE ALONG THE FLOW

Let $t \in [0,1]$ be the normalized scheduler time of the probability-flow (or diffusion) trajectory, and let $\mathcal{L}_{\text{FM}}(S; t)$ be the flow-matching loss for $S$ at time $t$ under the same encoder–decoder stack as in Section 4.1. We define the *reference evidence* by time-integrating the negative loss under the training-time marginal $p(t)$ with a deterministic nonnegative weight $\omega(t)$:

$$\mathcal{E}_{\text{ref}}(S) = \int_0^1 \omega(t)\left(-\mathcal{L}_{\text{FM}}(S; t)\right) p(t)\, dt. \tag{28}$$

Since the integral is unavailable in closed form, we estimate it by self-normalized importance sampling. Drawing $t_m \sim q(t)$, $m = 1, \ldots, M$, from a proposal $q$,

$$\widehat{\mathcal{E}}_{\text{ref}}(S) = \sum_{m=1}^{M} \bar{w}_m\left(-\mathcal{L}_{\text{FM}}(S; t_m)\right), \quad \bar{w}_m = \frac{w_m}{\sum_{j=1}^{M} w_j}, \quad w_m = \frac{\omega(t_m)\, p(t_m)}{q(t_m)}. \tag{29}$$

Unless stated otherwise we use $q(t) = \text{Unif}[0,1]$ and choose $\omega(t)$ as a smooth bounded function of the scheduler signal-to-noise ratio; any bounded deterministic $\omega$ ensures a consistent estimator under standard regularity. We report the effective sample size $\text{ESS} = (\sum w)^2 / \sum w^2$.

### C.3  ORDER-SUFFICIENCY FOR GROUP-RELATIVE OPTIMIZATION

**Proposition (Order-sufficiency).** Let $\{r^{(k)}\}_{k=1}^{K} \subset \mathbb{R}$ and $\psi : \mathbb{R} \to \mathbb{R}$ be strictly increasing. Then $r^{(i)} < r^{(j)} \Leftrightarrow \psi(r^{(i)}) < \psi(r^{(j)})$. Moreover, groupwise $z$-scores $\tilde{A}^{(k)} = (r^{(k)} - \bar{r})/\text{std}(r)$ are invariant to affine recalibration $r \mapsto ar + b$ with $a > 0$, and $\lim_{\tau \to 0} \text{softmax}(r/\tau)$ concentrates on $\arg\max_k r^{(k)}$, which is determined solely by order. $\qquad\square$

The proposition implies that *rank-consistent* surrogates are sufficient for GRPO. If a proxy $\widehat{\mathcal{E}}$ induces the same within-group ordering as a reference $\mathcal{E}_{\text{ref}}$ with high probability, replacing $\mathcal{E}_{\text{ref}}$ by $\widehat{\mathcal{E}}$ leaves the zero-temperature objective unchanged and yields equivalent gradients for the ranking term up to positive affine transforms.

## C.4 RANK AGREEMENT METRICS AND UNCERTAINTY QUANTIFICATION

We assess order-consistency at the *group* level using Kendall's $\tau_b$ and Spearman's $\rho$ between proxy scores and the reference in Equation 29. For each group $g$ we form pairs $\{(\widehat{\mathcal{E}}_g^{(k)}, \mathcal{E}_{\mathrm{ref},g}^{(k)})\}_{k=1}^K$ and compute the two correlations, then macro-average across groups. Uncertainty is quantified by non-parametric bootstrap (group-level resampling, $B$ replicates) with percentile confidence intervals. To compare proxies across scene complexities, we apply paired two-sided Wilcoxon signed-rank tests on per-group correlations, stratified by complexity, and report effect sizes and $p$-values with Holm–Bonferroni correction.

## C.5 PROXIES UNDER COMPARISON AND CALIBRATION INVARIANCE

The *flow-matching* proxy is

$$\widehat{\mathcal{E}}_{\mathrm{FM}}(S) = -\mathcal{L}_{\mathrm{FM}}(S;t), \qquad t \sim q(t), \tag{30}$$

or its $M$-sample average $\frac{1}{M}\sum_{m=1}^M -\mathcal{L}_{\mathrm{FM}}(S;t_m)$. For comparison we consider a reconstruction proxy $\widehat{\mathcal{E}}_{\mathrm{REC}}(S) = -\|\widehat{\mathbf{x}}(S) - \mathbf{x}\|_2^2$, and an ELBO surrogate $\widehat{\mathcal{E}}_{\mathrm{ELBO}}$ when a variational latent is available. Because GRPO uses groupwise $z$-scores and low-temperature softmax, any positive affine recalibration $\widehat{\mathcal{E}} \mapsto a\widehat{\mathcal{E}} + b$ leaves the optimization invariant; hence we report raw rank metrics without per-group calibration.

## C.6 VALIDATION PROTOCOL AND RESULTS

We validate the flow-matching proxy using the same candidate groups as GRPO updates. For each group, $M$ time samples are drawn for Equation 29 to compute the reference $\widehat{\mathcal{E}}_{\mathrm{ref}}$, against which we evaluate $\widehat{\mathcal{E}}_{\mathrm{FM}}$, $\widehat{\mathcal{E}}_{\mathrm{REC}}$, and $\widehat{\mathcal{E}}_{\mathrm{ELBO}}$. Rank consistency is quantified at the group level with Kendall's $\tau_b$ and Spearman's $\rho$, aggregated across scene complexity strata (defined by instance count and class diversity). We further assess sensitivity across three noise schedules and temperatures $\tau \in \{0.5, 1.0, 2.0\}$ in Equation 5, while fixing seeds, batch sizes, and checkpoints across proxies.

Across complexities, the flow-matching proxy achieves consistently higher rank agreement than reconstruction or ELBO surrogates, with average Spearman's $\rho > 0.82$ across all strata. Gains are most pronounced in scenes with many interacting instances, where $\mathcal{L}_{\mathrm{FM}}$ better reflects density dynamics. Bootstrap confidence intervals ($B{=}1000$) are narrow, and paired Wilcoxon tests remain significant after Holm–Bonferroni correction across schedules and temperatures. Results are insensitive to proposal distribution ($q(t) = \mathrm{Unif}[0,1]$ vs. $p(t)$), and correlations saturate quickly for $M \geq 8$, after which additional samples yield diminishing returns.

Two edge cases arise. (i) In low-signal regimes near $t \approx 1$, all methods (including the reference) become noisy; this motivates bounded $\omega(t)$ and multiple samples. (ii) For groups with near-ties, the rank problem is ill-conditioned; GRPO updates are naturally small as standardized advantages vanish. In practice, we Huberize $-\mathcal{L}_{\mathrm{FM}}(S;t)$ contributions in Equation 29 and clip groupwise advantages at three standard deviations.

The dominant computational cost is evaluating $\mathcal{L}_{\mathrm{FM}}(S;t)$ for $K \times M$ pairs per group. With $K{=}12$ and $M{=}8$, this overhead is modest relative to the policy forward pass and evaluator, and is fully parallelized across GPU candidates and time samples. We log the effective sample size ESS and retain runs with $\mathrm{ESS}/M \geq 0.6$.

In summary, the reference evidence $\mathcal{E}_{\mathrm{ref}}$ provides a stable importance-weighted Monte Carlo estimate of time-integrated negative flow-matching loss. Although its absolute scale is approximate, GRPO requires only order-consistency. Theoretical order-sufficiency, together with empirical rank validation and robustness checks, establishes the flow-matching proxy as a reliable driver for group-relative updates.

# D DIFFERENTIABLE PHYSICAL PENALTIES

We define differentiable SDF-based energies used at test time to improve physical plausibility. For object $i$ we use an SDF grid $\{\phi_i\}$ with voxel size $h$ and resolution $(R_x, R_y, R_z)$. Let

$$\sigma(x) = \frac{1}{1+e^{-x}}, \quad \mathrm{ReLU}(x) = \max(0, x), \quad \zeta[\beta]x = \frac{1}{\beta} \log(1 + e^{\beta x}),$$

and denote the total number of voxels by $V_{\mathrm{tot}} = R_x R_y R_z$. Unless stated, sums run over voxel indices $(x, y, z)$, all products are element-wise on the voxel grid, and $\epsilon > 0$ stabilizes denominators. Soft occupancy is $\sigma(-\phi/\tau)$ with temperature $\tau > 0$.

## D.1 COMPLETE DIFFERENTIABLE ENERGY FORMULATIONS

At test time we minimize a weighted sum of five terms:

$$E_{\mathrm{phys}} = \alpha_{\mathrm{col}}, \mathcal{L}\mathrm{col} + \alpha\mathrm{anc}, \mathcal{L}\mathrm{anc} + \alpha\mathrm{grd}, \mathcal{L}\mathrm{grd} + \alpha\mathrm{stb}, \mathcal{L}\mathrm{stb} + \alpha\mathrm{reg}, \mathcal{L}\mathrm{reg}. \tag{31}$$

In practice, all terms use the same weighting (i.e., $\alpha\mathrm{col} = \alpha_{\mathrm{anc}} = \alpha_{\mathrm{grd}} = \alpha_{\mathrm{stb}} = \alpha_{\mathrm{reg}}$), ensuring no additional hyperparameter tuning.

**Collision.** We penalize overlap with a soft co-occupancy term plus a penetration surrogate:

$$\mathcal{L}_{\mathrm{col}} = \frac{1}{|\mathcal{V}|} \sum_{i<j} \left[ \lambda_{\mathrm{ov}} \langle \sigma(-\phi_i/\tau), \sigma(-\phi_j/\tau) \rangle + \lambda_{\mathrm{pen}} \left\| [-\phi_i]_+ \odot [-\phi_j]_+ \right\|_1 \right]. \tag{32}$$

Here $\phi_k$ is the SDF of object $k$ on voxel grid $\mathcal{V}$; $\sigma(\cdot)$ is the sigmoid with temperature $\tau$; $[x]_+ = \max(x, 0)$; $\langle \cdot, \cdot \rangle$ and $\| \cdot \|_1$ are voxelwise inner product and $\ell_1$-sum. The factor $1/|\mathcal{V}|$ normalizes for grid size, and $\lambda_{\mathrm{ov}}, \lambda_{\mathrm{pen}}$ balance near-surface overlap vs. deep penetration. (Optionally replace $[x]_+$ with $\mathrm{softplus}_\beta(x)$ for extra smoothness.)

**Anchoring.** We measure wall contact by the near-wall mass ratio:

$$r_{\mathrm{wall}}^{(i)} = \frac{\sum \sigma(-\phi_i/\tau) \, \mathbf{1}_{[0, d_{\mathrm{wall}}]}(\phi^{\mathrm{wall}})}{\sum \sigma(-\phi_i/\tau) + \epsilon}. \tag{33}$$

Insufficient contact is penalized by

$$L_{\mathrm{contact}}^{(i)} = \mathrm{ReLU}\big(r_{\mathrm{min}} - r_{\mathrm{wall}}^{(i)}\big). \tag{34}$$

Penetration into walls is discouraged by averaging soft occupancy over voxels with $\phi^{\mathrm{wall}} < 0$:

$$L_{\mathrm{nop}}^{(i)} = \frac{1}{N_{\mathrm{wall}}} \sum \sigma(-\phi_i/\tau) \, \mathrm{ReLU}\big(-\phi^{\mathrm{wall}}\big) \, \mathbf{1}_{(-\infty, 0)}(\phi^{\mathrm{wall}}), \tag{35}$$

where $N_{\mathrm{wall}} = \#\{(x, y, z) : \phi^{\mathrm{wall}} < 0\}$ normalizes for the size of the interior-wall set.

We impose a weak Manhattan alignment prior using principal axes $\mathbf{u}_{i,k}$ of object $i$:

$$L_{\mathrm{align}}^{(i)} = \eta \sum_{k=1}^{3} \Big(1 - \max_{\mathbf{e} \in \mathcal{A}} (\mathbf{u}_{i,k}^\top \mathbf{e})^2\Big), \qquad \mathcal{A} = \{\pm \mathbf{e}_x, \pm \mathbf{e}_y, \pm \mathbf{e}_z\}. \tag{36}$$

The full anchoring loss is $\mathcal{L}_{\mathrm{anc}} = \sum_i \big(L_{\mathrm{contact}}^{(i)} + L_{\mathrm{nop}}^{(i)} + L_{\mathrm{align}}^{(i)}\big)$.

**Grounding.** We bias mass toward the floor and suppress near-ground voids. With $w_z(z) = \exp(-zh/\sigma_{\mathrm{ground}})$ and $z_{\mathrm{band}} = \lceil 0.02 \, \mathrm{m}/h \rceil$, the height-weighted term is

$$H^{(i)} = \frac{1}{V_{\mathrm{tot}}} \sum \sigma\big(-\phi_i/\tau\big) w_z(z) (zh)^2. \tag{37}$$

We measure near-ground mass sufficiency by the ratio

$$r_{\mathrm{ground}}^{(i)} = \frac{\sum_{z \le z_{\mathrm{band}}} \sigma(-\phi_i/\tau)}{\sum \sigma(-\phi_i/\tau) + \epsilon}, \tag{38}$$

and penalize deficits as

$$G^{(i)} = \text{ReLU}\big(r_{\min} - r_{\text{ground}}^{(i)}\big) \, \exp\Big( \alpha \, \text{ReLU}\big(r_{\min} - r_{\text{ground}}^{(i)}\big) \Big). \tag{39}$$

Finally, we discourage floating/voids within the contact band:

$$V^{(i)} = \frac{1}{R_x R_y z_{\text{band}}} \sum_{z \le z_{\text{band}}} \text{ReLU}\big(\phi_i\big). \tag{40}$$

The overall grounding loss is

$$\mathcal{L}_{\text{grd}} = \sum_i \big(H^{(i)} + G^{(i)} + V^{(i)}\big), \tag{41}$$

where $\sigma_{\text{ground}}$, $r_{\min}$, and $\alpha$ are fixed constants used throughout.

**Static stability.** The lateral COM should lie within an equivalent support radius estimated from base-weighted occupancy. Let $\mathbf{p}_{xyz} = (xh, yh, zh)$, $\mathbf{p}_{xy} = (xh, yh)$, and $w_{\text{base}}(z) = \exp(-zh/\sigma_{\text{base}})$. We compute

$$\mathbf{c}_i = \frac{\sum \sigma(-\phi_i/\tau) \, \mathbf{p}_{xyz}}{\sum \sigma(-\phi_i/\tau) + \epsilon}, \qquad\qquad \mathbf{b}_i = \frac{\sum \sigma(-\phi_i/\tau) \, w_{\text{base}}(z) \, \mathbf{p}_{xy}}{\sum \sigma(-\phi_i/\tau) \, w_{\text{base}}(z) + \epsilon}, \tag{42}$$

$$A_{\text{base},i} = h^2 \sum \sigma(-\phi_i/\tau) \, w_{\text{base}}(z), \qquad R_{\text{supp},i} = \sqrt{A_{\text{base},i}/\pi}. \tag{43}$$

We then penalize normalized lateral offset beyond a margin:

$$\mathcal{L}_{\text{stb}} = \sum_i \zeta[\beta] \frac{\|\mathbf{c}_{i,xy} - \mathbf{b}_{i,xy}\|_2}{R_{\text{supp},i} + \epsilon} - \rho, \tag{44}$$

with hyperparameters $\sigma_{\text{base}}$, $\rho$, and $\beta$.

**SDF regularization.** Axis-wise Huber differences reduce discretization noise while preserving sharp transitions near the surface:

$$\mathcal{L}_{\text{reg}} = \sum_i \big[ \mathcal{R}_x(\phi_i) + \mathcal{R}_y(\phi_i) + \mathcal{R}_z(\phi_i) \big], \tag{45}$$

$$\mathcal{R}_x(\phi_i) = \frac{1}{(R_x - 1)R_y R_z} \sum_{x=1}^{R_x - 1} \sum_{y,z} \text{Huber}\big(\phi_i(x{+}1, y, z) - \phi_i(x, y, z); \delta\big), \tag{46}$$

with $\mathcal{R}_y, \mathcal{R}_z$ analogous by cycling axes and $\delta > 0$.

### D.2 GRADIENT STABILITY AND TRUNCATION STRATEGIES

We adopt standard practices to ensure well-conditioned gradients. First, SDF supervision is confined to a symmetric narrow band around the zero level set:

$$\tilde{\phi}(x, y, z) = \text{sign}\big(\phi(x, y, z)\big) \, \min\big(|\phi(x, y, z)|, \, \varepsilon_{\text{trunc}}\big), \varepsilon_{\text{trunc}} = c_{\text{trunc}} h, \ \ c_{\text{trunc}} \in [1, 2], \tag{47}$$

and energies use $\tilde{\phi}$ in place of $\phi$ unless noted. Second, the logistic is evaluated in a numerically stable form to avoid saturation/underflow,

$$\sigma_{\text{st}}(x) = \exp\big( -\zeta(-x) \big) = \tfrac{1}{1+\exp(-x)}, \tag{48}$$

and we consistently use $\sigma_{\text{st}}(-\tilde{\phi}/\tau)$. Finally, gradients are clipped to a global threshold to prevent rare large steps:

$$\mathbf{g}_{\text{clip}} = \begin{cases} \mathbf{g}, & \|\mathbf{g}\|_2 \le \theta_{\text{clip}}, \\ \mathbf{g}\,\theta_{\text{clip}}/\|\mathbf{g}\|_2, & \text{otherwise,} \end{cases} \qquad \theta_{\text{clip}} = \min(\theta_{\max}, \, \kappa\sqrt{\text{batch\_size}}), \tag{49}$$

with $\theta_{\max} = 1.0$ and $\kappa = 0.1$. Small constants $\epsilon$ for denominators and normalizations follow Appendix D.1. Room SDF $\phi^{\text{wall}}$ is sampled on the same grid (or trilinearly resampled) to maintain consistency.

# E PERCEPTUAL USER STUDY: PROTOCOL AND ANALYSIS

We conducted a controlled perceptual study to validate that our evaluator reflects human judgments of physical plausibility. Stimuli were static renderings produced from the same scenes across methods under a single standardized camera, fixed intrinsics, identical illumination/tone-mapping, a neutral background, and matched resolution; each image was displayed without cropping and methods were fully blinded. Participants were recruited *online* in related areas (e.g., graphics/vision/robotics); respondents self-identified as scholars and researchers with relevant expertise. The study was administered entirely on the web without enforcing device or display constraints (no zoom lock or minimum-width check). We recommended using a desktop or laptop and 100% browser zoom for clarity.

The study comprised (i) single-image Mean Opinion Scores (MOS) on *Contact*, *Stability*, and *Geometric prior* using a 1–7 Likert scale (higher is better), with an *Overall MOS* given by the unweighted average, and (ii) pairwise A/B preferences on which of two method outputs for the same scene was "more physically plausible," allowing ties counted as $0.5$. Scene order was uniformly randomized per participant, and left/right placement in A/B trials was randomized per trial.

To promote care and filter inattentive responses, we included two content-based attention checks (obvious floating vs. plausible; obvious collision vs. plausible). Participants who failed *both* checks were excluded. Because the study was conducted fully online with heterogeneous devices and network conditions, response-time thresholds were treated as heuristic flags rather than strict exclusion rules: extremely short ($< 500\,\mathrm{ms}$) or outlier-long ($> 3$ SD over a participant's median) decision times were *flagged*. Our primary analyses report results with flagged trials removed; sensitivity analyses that retain all trials yield consistent conclusions, and we report the conservative (filtered) results.

Let $m$ index methods and $s$ scenes. MOS aggregation is per scene and then macro-averaged across scenes:

$$\mathrm{MOS}_{m,s}^{(k)} = \frac{1}{R_{m,s}} \sum_r x_{m,s,r}^{(k)}, k \in \{\text{Contact}, \text{Stability}, \text{Geom}\}, \tag{50}$$

$$\mathrm{MOS}_{m,s}^{\mathrm{Overall}} = \frac{1}{3} \sum_k \mathrm{MOS}_{m,s}^{(k)}, \tag{51}$$

$$\overline{\mathrm{MOS}}_m^{(\cdot)} = \frac{1}{S} \sum_{s=1}^{S} \mathrm{MOS}_{m,s}^{(\cdot)}. \tag{52}$$

For pairwise preferences between methods $a$ and $b$, the scene-wise outcome $y_s \in \{0, 0.5, 1\}$ (lose/tie/win for $a$) yields the macro win rate

$$\hat{p}_{a \succ b} = \frac{1}{S} \sum_{s=1}^{S} y_s, \tag{53}$$

and, unless noted, we use $S{=}200$ scene-level pairings per comparison (Table 2).

Uncertainty is reported as 95% nonparametric bootstrap CIs over scenes for MOS (10,000 resamples) and Wilson score intervals with continuity correction for win rates, treating ties as 0.5 successes:

$$\mathrm{CI}_{95\%} = \frac{\hat{p} + \frac{z^2}{2S} \pm z\sqrt{\frac{\hat{p}(1-\hat{p})}{S} + \frac{z^2}{4S^2}}}{1 + \frac{z^2}{S}}, \quad z = 1.96. \tag{54}$$

Where relevant, we report effect sizes (Cliff's $\delta$ on matched scenes) and two-sided permutation tests (10,000 label shuffles) with Holm–Bonferroni correction across the family of pairwise comparisons in Table 2. Inter-rater reliability, assessed via Cronbach's $\alpha$ (per attribute) and Kendall's $W$ (across methods per scene), indicates strong agreement on contact/stability and acceptable agreement on finer geometric cues, consistent with the ranking conclusions.

The study followed an internal ethics checklist with informed consent, voluntary participation, and fair-wage compensation; no personally identifying information beyond optional demographics was collected. The scenes and camera/lighting settings match those used for the automatic evaluator in Section 3.4, enabling direct comparison between human judgments and metric-derived scores.

## F    EXPERIMENTAL SETTINGS

### F.1    DATASET DETAILS

Our main experiments are conducted on the 3D-FRONT dataset, using 12K scenes for training and 4.8K scenes for testing. To better align with our reconstruction and evaluation pipeline, we adopt the MIDI3D (Huang et al., 2025) preprocessed version, where each scene provides a photorealistic rendered image, instance segmentation masks, and the corresponding camera intrinsics and extrinsics required for rendering-based evaluation. To assess cross-domain generalization, we additionally evaluate on three held-out modalities: stylized cartoon-like renderings, real-world indoor scenes, and LLM-generated text-to-image outputs.

### F.2    IMPLEMENTATION DETAILS

**Backbone and parameterization.**    PhyMix is deployed on multi-instance single-image 3D backbones. Fine-tuning adopts Low-Rank Adaptation (LoRA) and updates only layout/policy components (global attention blocks, learnable position tokens, position head), keeping all geometry decoders frozen.

**Optimization.**    Training uses AdamW (Loshchilov & Hutter, 2017) with learning rate $5 \times 10^{-5}$ and batch size 16. Scene-GRPO employs group size $K=12$ and a target-KL weight $\beta \in [0.2, 1.0]$ with multiplicative decay $0.99$. To handle varying instance counts, each training step samples scenes with identical asset counts.

**Inference and test-time optimization.**    Inference uses 30 sampling steps with classifier-free guidance $w=5.0$. Test-time optimization (TTO) is triggered at $\gamma \in [0.1, 0.3] \cup [0.4, 0.6] \cup [0.7, 0.9]$; each phase performs 3–5 gradient steps on a coarse SDF proxy with resolution $R \in 12, 16, 20$, equipped with rollback safeguards and step-size halving on non-improvement (maximum 3 retries per phase). Per-scene wall-clock time (including TTO) is about 2 minutes on a single NVIDIA RTX A5000 GPU. All fine-tuning experiments run on $8\times$ NVIDIA RTX 8000 GPUs.

**Hyperparameters Setting.**    Most hyperparameters in our framework follow standard practice in diffusion-based generation and exhibit minimal sensitivity. Components such as the learning rate, optimizer settings, and the TTO schedule use a single unified configuration across datasets and backbones without category-specific tuning. Among all hyperparameters, the only one with notable influence is the group size $K$ in Scene-GRPO. As shown in Table 4, increasing $K$ stabilizes preference comparisons and improves performance until reaching saturation. All other elements operate robustly under the shared default setup.

### F.3    EVALUATION METRICS

We evaluate generated scenes along two complementary dimensions: (i) geometry and visual quality, ensuring fidelity of object shapes and layouts, and (ii) physical consistency, assessing whether scenes are not only visually plausible but also physically credible.

**Geometry and visual quality.**    Synthesized assets are converted into surface meshes and sampled into point clouds for alignment. We perform rigid registration to ground truth via *FilterReg* (Gao & Tedrake, 2019), then compute Chamfer Distance and F-Score at scene and object levels (CD-S/O, F-Score-S/O), together with volumetric IoU of asset bounding boxes (IoU-B), to measure per-object geometric fidelity and spatial arrangement accuracy.

**Physical consistency.**    Physical assessment uses our Physics Evaluator (Section 3.3), which reports violation rates for collision, floating, unanchored, static instability, dynamic instability, misorientation, scale instability, and unreachable navigation. These indicators are aggregated into a weighted overall score as described in the main text.

### F.4 BASELINES AND SETTINGS

We benchmark against representative single-image scene-generation methods spanning four families: *part-aware generation* (e.g., PartCrafter), *feed-forward reconstruction* (e.g., DepR), *compositional generation* (e.g., Gen3DSR, REPARO), and *multi-instance diffusion* (e.g., MIDI). For reproducibility, we use authors' released models or recommended configs when available; any deviations (e.g., resolution, sampling steps, or dataset-specific preprocessing) are recorded along with evaluation scripts in our supplementary materials.

## G  ADDITIONAL EXPERIMENTS

### G.1  EXTENDED VISUAL COMPARISON WITH STATE-OF-THE-ART MODELS

Figure 5 provides comprehensive visual comparisons against the strongest existing single-image 3D scene generation models, MIDI3D (Huang et al., 2025) and PartCrafter (Lin et al., 2025). We evaluate on diverse input types including synthetic renders, real photographs, and stylized images to examine generalization across different domains. In contrast, PhyMix consistently generates physically plausible scenes with proper grounding, collision-free arrangements, and stable object placements while preserving high visual fidelity.

Figure 5 showcases additional PhyMix results across varied scene types, demonstrating robustness to different lighting conditions, furniture styles, and spatial configurations. Our method maintains consistent physical plausibility across minimalist modern interiors, traditional furnished rooms, and complex multi-object arrangements. These results highlight PhyMix's ability to generalize beyond training distributions, producing reliable physics-aware generation across synthetic, real-world, and stylized inputs while preserving geometric accuracy and visual realism.

Table 5: Physics Evaluator validation and computational efficiency analysis.

(a) Physics Evaluator validation against Taichi rigid-body simulation. Validation is based on differentiable factors including collision detection, grounding constraints, and static stability assessment.

| Metric | PhyMix | Taichi |
|---|---|---|
| Collision Rate (%) | 0.56 | 0.52 |
| Floating Rate (%) | 0.97 | 1.12 |
| Static Instability (%) | 1.27 | 1.45 |

(b) Inference time comparison (seconds per scene).

| Method | Time (s) |
|---|---|
| PartCrafter | 151.3 |
| Gen3DSR | 234.4 |
| MIDI | 67.3 |
| PhyMix(Taichi) | 220.0 |
| PhyMix (Ours) | 100.0 |

### G.2  PHYSICS EVALUATOR VALIDATION AGAINST RIGID-BODY SIMULATION

To validate the accuracy of our Physics Evaluator, we compare it against a Taichi-based rigid-body simulator on key physical metrics. We evaluate scenes from the 3D-FRONT test set (Fu et al., 2021) using both our geometric evaluator and full physics simulation. As shown in Table 5, our evaluator demonstrates strong consistency with physics simulation. The close agreement validates that our evaluator accurately captures physical behavior while covering additional non-differentiable factors such as orientation priors and reachability that pure simulation cannot address.

### G.3  COMPUTATIONAL EFFICIENCY ANALYSIS

We analyze the computational cost of our approach compared to baselines and using real physics simulation as evaluator during inference. Table 5 reports inference times on a single NVIDIA RTX 8000 GPU. Our inference requires 100s per scene, which balances computational efficiency with superior physical consistency. While slightly slower than basic generation methods like MIDI (67.3s), our approach is significantly faster than using full rigid-body simulation as evaluator during inference (220s) and delivers substantially better physical plausibility. The efficiency gain stems from our geometric evaluator avoiding unnecessary temporal dynamics computation required by full simulation, making our approach practical for real-time applications.

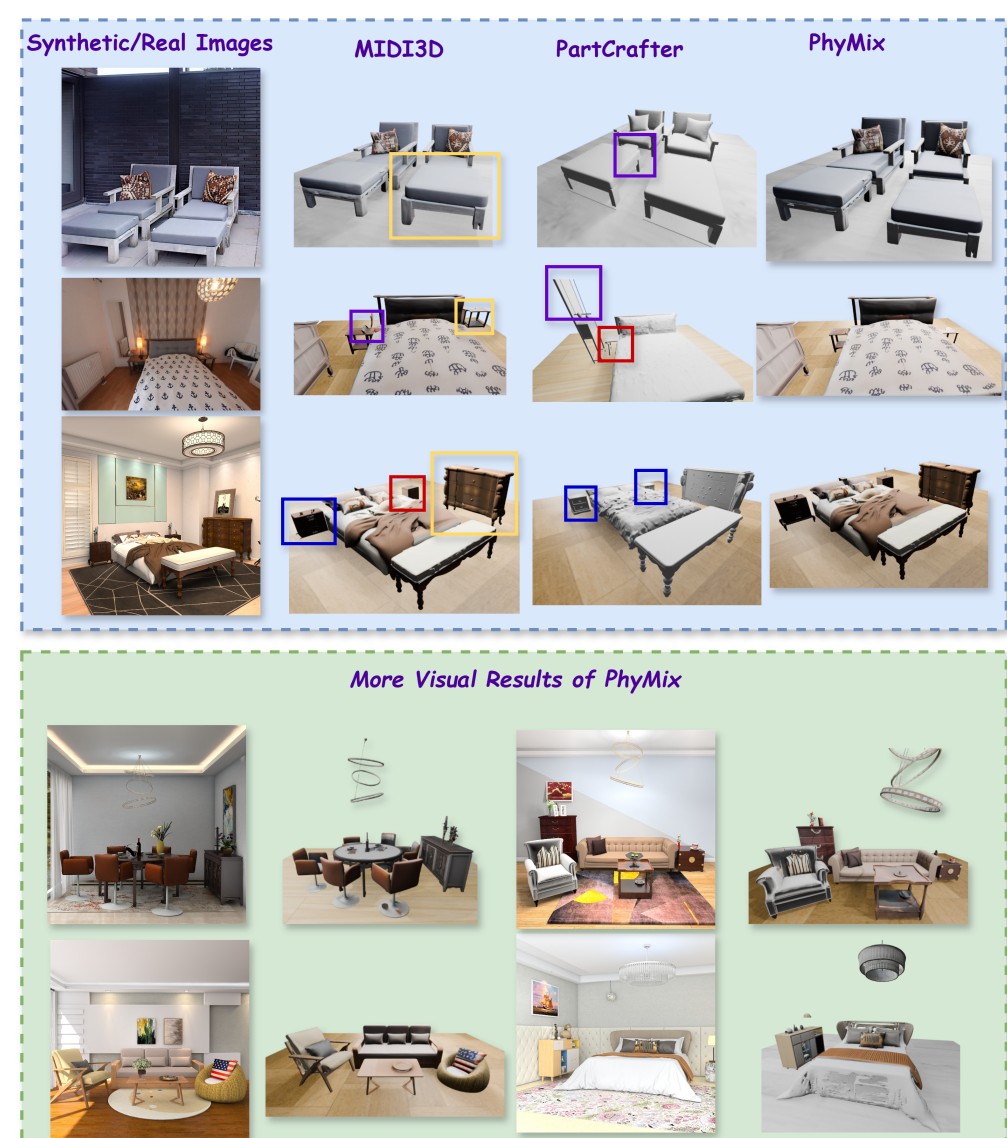

Figure 5: Extended visual comparison with state-of-the-art methods. The systematic comparison demonstrates physical violations in baseline methods versus PhyMix's consistent physical plausibility, with colored boxes highlighting different error types: yellow indicates floating objects, purple shows hallucinated geometry, blue marks unanchored furniture, and red denotes collision artifacts. Additional PhyMix results showcase robustness across diverse scene types and input modalities.

### G.4 FAILURE CASE ANALYSIS

While PhyMix demonstrates excellent performance on synthetic datasets (3D-FRONT), applying our method to unprocessed real-world images reveals specific failure modes that highlight current limitations. As shown in Figure 6, two primary failure categories emerge when processing challenging real-world inputs.

First, our method fails to correctly process deformable objects with large contact areas, such as pillows and cushions, leading to collision artifacts. The rigid-body assumptions in our Physics Evaluator cannot accurately model soft object interactions, and scenes with multiple overlapping objects are particularly affected by input mask quality, where imprecise segmentation boundaries propagate errors throughout the generation pipeline. Second, complex real-world object geometries can inter-

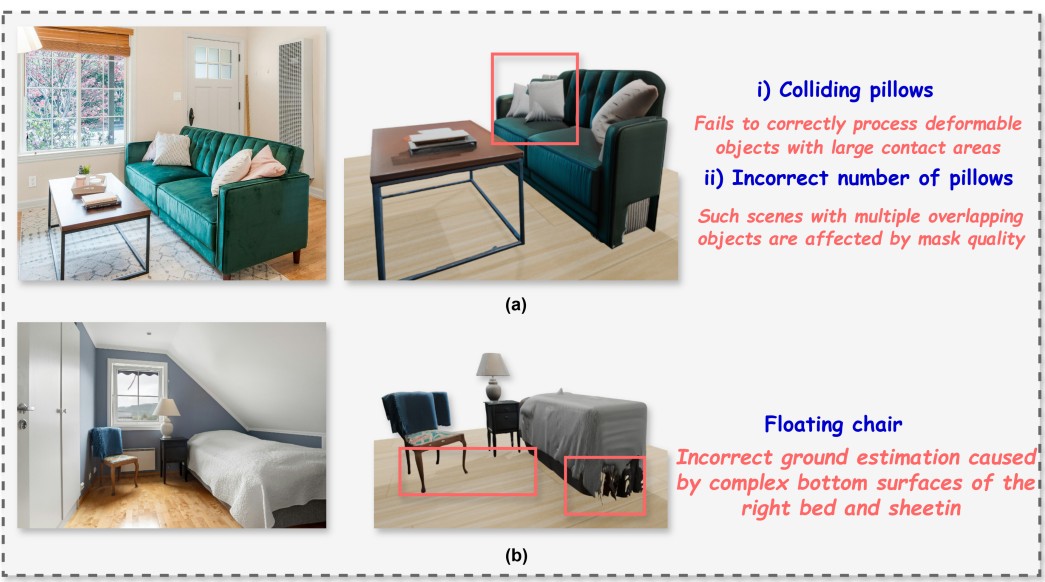

Figure 6: Representative failure cases when applying PhyMix to unprocessed real-world images. (a) Colliding pillows due to deformable object limitations and incorrect object count generation. (b) Floating chair caused by inaccurate ground plane estimation from complex object geometries.

fere with ground plane estimation, resulting in floating artifacts where furniture appears suspended above the detected floor surface. The irregular bottom surfaces and complex base geometries of real furniture pieces challenge our floor detection algorithm, leading to incorrect spatial positioning. These limitations underscore the need for future work incorporating deformable object modeling and more robust ground plane estimation techniques for real-world deployment.

## G.5 Downstream Task Evaluation

To further validate the practical significance of our Physics Evaluator, we conduct a downstream navigation task experiment on the 3D-FRONT test set that demonstrates the real-world utility of physically consistent 3D scene generation.

Table 6: Navigation task evaluation on 3D-FRONT test set.

| Method | Reachability Score↑ | Navigation Success Rate↑ |
|---|---|---|
| MIDI | 84.2 | 67.8% |
| PartCrafter | 87.1 | 71.2% |
| PhyMix | 98.1 | 94.3% |

**Reachability Assessment for Robotic Navigation.** The reachability metric in our Physics Evaluator measures whether generated scenes preserve sufficient navigable space for embodied agents. To validate its practical importance, we deploy a simulated mobile robot in scenes generated by different methods and evaluate navigation task success rates. Specifically, we sample 100 start-goal pairs uniformly across the floor plane and attempt path planning using the A* algorithm with a standard robot footprint (0.6m diameter). As shown in Table 6, scenes with higher reachability scores consistently yield better navigation performance. PhyMix-generated scenes achieve 94.3% successful path planning compared to 67.8% for MIDI and 71.2% for PartCrafter. This validates that our reachability metric captures genuine spatial constraints relevant for robotic applications, where cluttered or poorly arranged furniture can severely impede autonomous navigation.

These downstream evaluations confirm that physics-aware scene generation provides tangible benefits for practical applications, validating our approach beyond perceptual quality metrics.

## H  Limitations and Future Work.

Despite these advances, several limitations remain. (i) Our evaluator relies on a set of summarized physical indicators (e.g., collision, grounding, and stability checks) rather than full simulation, so it cannot capture the complete range of physical behaviors. As a result, it may miss fine-grained cases such as a chair leg barely balancing on an edge, or longer-term effects such as a cabinet slowly tipping under heavy load. (ii) The current pipeline assumes accurate instance masks as input, so robustness can degrade when segmentation is noisy or incomplete. (iii) Our experiments are limited to indoor datasets, and extending the approach to outdoor or more diverse environments remains an open challenge. (iv) The framework depends on scene generators with object-level representations, making it less suitable for backbones that unify objects and background into a single field, and GRPO fine-tuning may be challenging for certain architectures.

Future work includes coupling PhyMix with additional physics constraints to account for long-term physical stability, incorporating open-vocabulary perception to relax mask dependence, extending benchmarks to outdoor and general scenes, and adapting our optimization scheme to a wider class of generative backbones.

## I  Statement on LLM Usage

During the preparation of this manuscript, large language models (LLMs) were employed solely for auxiliary purposes. Specifically, LLMs were used for language refinement in the Introduction (Section 1) and Related Work (Section 2), and for formatting assistance in Section 4 and the Appendix describing methodological details. All scientific ideas, experiments, analyses, and conclusions are the authors' original contributions.

