# OpenReview forum: "PhyMix: Towards Physically Consistent Single-Image 3D Indoor Scene Generation with Implicit–Explicit Optimization"
_ICLR.cc/2026/Conference — ICLR 2026 Conference Withdrawn Submission_

### Official Review · Reviewer_TRuL · 2025-10-21

**Soundness:** 2
**Presentation:** 1
**Contribution:** 1
**Rating:** 2
**Confidence:** 3

**Summary:**

The paper proposes a so-called physics evaluator for evaluating the quality of image-example based indoor scene synthesis. Based on the proposed evaluator functions, the authors integrate them into a policy learning model and a test-time optimizer for implicit learning and explicit refinement. Experiments show that their proposed method performs better than SOTA models in terms of Chamfer distance, F-Score and FID on the 3D-FRONT test set.

**Strengths:**

The authors well-analyzed the physics-related aspects such as contact (grounding, collision-free), stability etc among existing models, and pointed the significance of incorporating these essences into establishing a more profound image-to-3D-scene model.

**Weaknesses:**

1. The paper is very hard to follow. One of the key contribution of the paper, the evaluator metrics, is not given priority in the main paper (L195-L200), but instread appeared in Supp with complicated mathematic definitions. Throughout the paper, too much symbols shown without clear explanations.
2. Some contents are not described constant throughout the paper. E.g. The input-output interface is not clearly explained or not consistent. In Figure 2, it shows the inputs are the RGB image and instance masks. However, in SubSection5.1, there is no description of how to work on the 3D-FRONT test set. As far as I know, there is no provided such RGB images and instance masks.
3. There is no comparison between the proposed policy learning model and a test-time optimizer with exisiting models. I'm curious about what is the key essence that improves the model to a great extent.
4. Visual results are not enough. There are only 4 examples shown in Figure 4. I'd suggest to provide more visual comparison results.
5. No visual ablation study shown in Supp.
6. Computational cost is not provided.
7. Failure cases are suggested to be added.
8. It would be great to provide some downstream tasks to enrich the experimental setup.

**Questions:**

See weakness

---

> ### Author Response · Authors · 2025-11-15
> **Response to Reviewer TRuL**
>
> We would like to thank the reviewers for recognizing the strengths of PhyMix, particularly our analysis of physics-related aspects such as contact, grounding, collision-free placement, and stability, and for providing detailed and constructive feedback. Below, we respond to each point individually.
>
> ### **On Weaknesses**
>
> ---
>
> ***“The paper is very hard to follow and too many symbols are shown without clear explanations.”***
>
> We thank the reviewer for highlighting the clarity issue regarding the Physics Evaluator. Our paper follows a concise four-step logic:
>
> 1. Identify the problem. We analyze existing single-image 3D scene generators and show that they frequently violate basic physical constraints (Sec. 3.1–3.2).
>
> 2. Define a unified Physics Evaluator. We introduce four aspects (Geometric Priors, Contact, Stability, Deployability) and nine sub-indicators that collectively capture object-level, inter-object, scene-level, and task-level plausibility (Sec. 3.3).
>
> 3. Validate the evaluator. We run a user study and show strong correlation between evaluator scores and human judgments (Sec. 3.4).
>
> 4. Use evaluator feedback to improve generation. We integrate evaluator signals into training (Scene-GRPO) and inference (TTO) to correct physical errors in the generated scenes (Sec. 4).
>
> This provides a clean conceptual flow from diagnosing the problem, to defining measurable evaluation criteria, to using these criteria to improve generation quality.
>
> To address the reviewer’s concern about unclear notation, we explicitly list the key symbols used in the Physics Evaluator and their roles:
>
> | Symbol | Meaning | Role in Evaluator |
> |--------|---------|-------------------|
> | `S` | The full 3D scene state containing all objects | Input to all evaluator modules |
> | `(R_i, t_i)` | Rotation matrix and translation of object *i* | Defines each object’s 6-DoF pose |
> | `g_i` | Geometric prior validity of object *i* (uprightness, scale, gravity alignment) | Determines whether object-level priors are satisfied |
> | `c_{ij}` | Contact / collision indicator between objects *i* and *j*, or between an object and the floor | Used to detect floating, penetration, and support relations |
> | `s_i` | Stability score of object *i* based on center-of-mass and support geometry | Determines static and short-horizon dynamic stability |
> | `d(x)` | Deployability field over spatial position *x* indicating free navigable regions | Used to evaluate path reachability and free-space usability |
> | `K` | Group size in Scene-GRPO | Only hyperparameter with significant influence in sensitivity studies |
>
> In the revised version, we are rewriting Section 3.3, summarizing the inherent connections between evaluation indicators and clearly defining sub-indicators directly in the main text.
>
> ---
>
> ***“As far as I know, there is no provided such RGB images and instance masks.”***
>
> The experiments on 3D-FRONT are conducted using the preprocessed version of the dataset, available at: https://huggingface.co/datasets/huanngzh/3D-Front. We directly follow their preprocessing setup, which provides rendered RGB images, instance masks, and aligned 3D models derived from the official 3D-FRONT assets. This pipeline has been widely adopted in recent works, including SceneGen [1] and PartiCraft [2].
> We are modifying parts of the dataset to avoid ambiguity in Appendix F.
>
> **References**
>
> [1] Yanxu Meng, Haoning Wu, Ya Zhang, and Weidi Xie. *Scenegen: Single-image 3D scene generation in one feedforward pass*. arXiv:2508.15769, 2025.
> [2] Yuchen Lin, Chenguo Lin, Panwang Pan, Honglei Yan, Yiqiang Feng, Yadong Mu, and Katerina Fragkiadaki. *PartCrafter: Structured 3D mesh generation via compositional latent diffusion transformers*. arXiv:2506.05573, 2025.

---

> ### Author Response · Authors · 2025-11-16
> **Response to Reviewer TRuL (continued under Weaknesses)**
>
> **On Weaknesses**
>
> ---
>
> ***There is no comparison between the proposed policy learning model and a test-time optimizer with existing models, what is the key essence that improves the model to a great extent.***
>
> First, this work aims to fill a gap that has not been addressed in prior literature: existing single-image 3D scene generators lack a complete and standardized way to measure physical plausibility. Current methods include only isolated cues such as collision or grounding [1,2,3], which cannot capture stability, geometric priors, or deployability. The unified physics evaluator proposed in this work fills this gap and provides the first comprehensive protocol needed for fair and thorough evaluation of physical consistency. Furthermore, our method, based on the results, is superior to existing single-image indoor baseline generation methods.
>
> Second, prior test-time optimization strategies are essentially patch-based. They apply a few differentiable penalties as temporary post-processing and were never designed as standalone optimization frameworks with any training component. Because these TTO heuristics operate only after sampling and only on partial constraints, they cannot define a coherent baseline that can be meaningfully compared with a full training-time optimization method. Therefore, they are not included as direct baselines.
>
> Finally, the proposed preference-based GRPO framework fundamentally subsumes the role of such TTO refinements. By optimizing the sampling distribution using physics-guided preference signals, the model learns both differentiable and non-differentiable constraints during training rather than relying on post-hoc corrections. This enables our method to absorb the benefits of TTO while allowing the model to genuinely learn physically correct indoor layouts, thereby explaining the substantial improvements observed in our experiments.
>
> We are actively adding visual contrasts, downstream tasks, ablation and failure cases, as well as reorganizing the symbol definitions in the article to make the results more complete and convincing.
>
> ---
>
> **References**
>
> [1] Yanxu Meng, Haoning Wu, Ya Zhang, and Weidi Xie. Scenegen: Single-image 3D scene generation in one feedforward pass. arXiv:2508.15769, 2025.
>
> [2] Kaixin Yao, Longwen Zhang, Xinhao Yan, Yan Zeng, Qixuan Zhang, Lan Xu, Wei Yang, Jiayuan Gu, and Jingyi Yu. CAST: Component-aligned 3D scene reconstruction from an RGB image. TOG, 2025.
>
> [3] Yandan Yang, Baoxiong Jia, Peiyuan Zhi, and Siyuan Huang. PhyScene: Physically interactable 3D scene synthesis for embodied AI. CVPR, 2024.

---

> ### Author Response · Authors · 2025-11-27
> **Follow-up Response for Reviewer TRuL**
>
> Dear Reviewer TRuL：
>
> Thank you again for your detailed review and for identifying clarity issues and missing experimental elements. We have carefully revised the manuscript according to your comments.
>
> We have rewritten the Physics Evaluator section to improve readability and placed key definitions and symbol descriptions directly in the main text (**Section 3.3**), rather than relying on supplemental material.
>
> Dataset usage and input–output interfaces have been clarified, including how RGB images and instance masks are obtained for 3D-FRONT (**Section 5.1**, with details in **Appendix F**).
>
> Additional visual comparisons, more qualitative results, and failure cases have been added to strengthen empirical evidence (**Section 5.4**, extended in **Appendix G**).
>
> Computational cost and downstream task demonstrations have been included to address practicality and usage scope (**Section 5.4** and **Appendix G**).
>
> As the discussion stage is approaching its end, we would be grateful if you could take a moment to review these updates or let us know if any further clarification is needed. Your comments have helped significantly improve readability, completeness, and evaluation depth of this work.
>
> Best regards,
> Authors of Submission 968

---

### Official Review · Reviewer_5eD1 · 2025-10-29

**Soundness:** 3
**Presentation:** 3
**Contribution:** 3
**Rating:** 6
**Confidence:** 4

**Summary:**

This paper proposes *PhyMix*, a physics-guided framework for single-image 3D indoor scene generation. The authors introduce a unified *Physics Evaluator* consisting of four aspects and nine measurable physical constraints, and integrate its feedback into both training (implicit preference alignment using Scene-GRPO) and inference (explicit Test-Time Optimization). Experiments on 3D-FRONT show consistent improvements in physical plausibility and geometric fidelity, with qualitative results across various image domains.

**Strengths:**

1. The proposed Physics Evaluator provides a comprehensive and unified measurement of physical consistency, covering contact, stability, geometric priors, and deployability.
2. The combination of implicit optimization (Scene-GRPO) and explicit refinement (TTO) is conceptually elegant and appears effective in improving physical plausibility.
3. The method generalizes to multiple input domains (real, synthetic, cartoon, LLM-generated), showing robustness and practical applicability.
4. The paper is overall well-written and should be easy to follow.

**Weaknesses:**

1. The training pipeline depends on the Physics Evaluator, and some evaluation components (especially simulation-based stability $P_{sim}$ ) can be computationally expensive. The paper lacks a clear comparison of training/inference time and compute cost relative to baselines.
2. The Physics Evaluator contains many hyperparameters (as discussed in the appendix). It is not clear whether these hyperparameters are object-category dependent, or how sensitive the evaluator is to different scene compositions. More justification on robustness across object types is needed.
3. Qualitative comparisons in the main paper are limited. Given that physical consistency often manifests in motion or interaction, videos could better reflect physical plausibility. Currently, no supplementary video materials are provided.

**Questions:**

In Table 2, bolding the best results would make it easier for readers to compare methods.

---

> ### Author Response · Authors · 2025-11-15
> **Response to Reviewer 5eD1**
>
> We would like to thank the reviewer for highlighting the strengths of PhyMix, including its unified Physics Evaluator, the effectiveness of combining Scene-GRPO with TTO, its robustness across diverse input domains, and the overall clarity of the paper. Below, we respond to each weakness and question individually.
>
> **On Weaknesses**
>
> ---
>
> ***The training pipeline depends on the Physics Evaluator, and some evaluation components may be computationally expensive.***
>
> The overall training and inference cost remains within 1.2–1.5× of the backbone (MIDI3D [1]), concretely, MIDI3D’s per-scene inference typically takes around 70 seconds under 30 denoising steps, while our method (including TTO) takes around 100 seconds on the same hardware (RTX 8000). And we are reporting precise runtime numbers in the revision. We also include a small compute comparison table to clearly show the cost relative to baseline models.
>
> ---
>
> ***The Physics Evaluator contains many hyperparameters, and robustness across object types is unclear.***
>
> Most of the hyperparameters used in the evaluator are standard choices that follow common practice in diffusion-based generation and do not require careful tuning. These include the learning rate, basic optimizer settings, and the simple test-time optimization schedule used during inference. They are not designed with category-specific logic and have an insignificant influence on the final outcomes.
>
> Among all hyperparameters, the group size K in Scene-GRPO is the only one that shows a clear impact in our sensitivity study, where increasing K consistently improves physical plausibility until performance saturates, as shown in Table 4c. All other components operate reliably under a single unified configuration, and varying them within reasonable ranges produces only minor changes in violation rates without affecting the relative ranking of methods. We are updating Appendix F and Appendix G to present this shared configuration more clearly and to avoid potential ambiguity.
>
> ---
>
> ***Qualitative comparisons are limited, and no video results are provided.***
>
> We appreciate the suggestion. Our task focuses on generating static, physically plausible scenes from a single image, so visual quality is primarily reflected in geometric consistency rather than motion. Nonetheless, we are adding more qualitative comparisons and visual ablations in the supplementary material to better illustrate the improvements brought by GRPO and TTO.
>
> ---
>
> **Questions**
>
> ---
>
> ***In Table 2, bolding the best results would make comparisons clearer.***
>
> Agreed. We are improving the formatting of Table 2 and highlighting the best results to make comparisons more readable.
>
> ---
>
> **References**
>
> [1] Zehuan Huang, Yuan-Chen Guo, Xingqiao An, Yunhan Yang, Yangguang Li, Zi-Xin Zou, Ding Liang, Xihui Liu, Yan-Pei Cao, and Lu Sheng. MIDI: Multi-instance diffusion for single image to 3D scene generation. CVPR, pp. 23646–23657, 2025.

---

### Official Review · Reviewer_n4Rf · 2025-11-01

**Soundness:** 2
**Presentation:** 3
**Contribution:** 3
**Rating:** 6
**Confidence:** 4

**Summary:**

This paper proposes PhyMix, a framework for improving physical consistency in single-image 3D indoor scene generation. It introduces a Physics Evaluator that measures four aspects—contact, stability, geometric priors, and deployability—and uses these as rewards in a critic-free Group Relative Policy Optimization (Scene-GRPO) scheme. At inference, a Test-Time Optimizer (TTO) further refines object poses to remove collisions. Experiments on 3D-FRONT show PhyMix significantly improves physical realism without sacrificing visual quality.

**Strengths:**

• Physical consistency in 3D scene generation is a well-discussed problem, but most prior works tackle it in isolated ways. This paper takes a systematic and unified perspective to address it, which is valuable.
•  The idea of applying reinforcement learning (Scene-GRPO) to improve physical realism is well-motivated, showing a good balance between theory and practicality.

**Weaknesses:**

• The paper decomposes physical consistency into four main aspects (contact, stability, geometric priors, deployability) and nine sub-metrics, which is conceptually clear but lacks theoretical or empirical justification for this particular taxonomy. Similar attempts have been made in recent works such as LayoutDreamer [Zhou et al., 2025], which also enforces physical plausibility through contact and penetration constraints in text-to-3D scene generation. The paper would benefit from a clearer explanation of why these four aspects are chosen, how they interact, and whether they comprehensively cover all major physical inconsistencies.

• Although the reported results show improvement in physical realism, the experiments are primarily limited to 3D-FRONT, with only a few scene variations. There is no cross-domain validation (e.g., cluttered indoor scenes) or analysis on unseen object categories. Furthermore, ablation on each sub-metric of the Physics Evaluator is missing, leaving it unclear which aspects contribute most to the gains.

• The proposed Group Relative Policy Optimization (GRPO) is an interesting critic-free RL approach, but its advantages over standard algorithms such as PPO [Schulman et al., 2017] or AWR [Peng et al., 2019] are not sufficiently quantified. It would strengthen the claim if the authors could compare sample efficiency, stability, or convergence behavior to existing RL baselines.

• Several parts (e.g., Section 3.1 on scene representation, Equation definitions of the Physics Evaluator) are written only in prose without explicit symbols or equations, which weakens reproducibility. Moreover, the gradient flow between the evaluator, GRPO, and generator is not fully detailed—does the Physics Evaluator participate in backpropagation, or only as a reward signal? Clarifying this would improve methodological transparency.

**Questions:**

1. How did you determine the weights for the nine sub-metrics? Do these weights transfer across datasets/tasks without retuning? If not, do you have any results on automated weight selection or a sensitivity analysis to support stability and reuse?

2. Compared with differentiable physics or analytic constraints, how does your method differ in sample efficiency, convergence speed, and failure modes? Do you have any small-scale real-world or sim-to-real results to demonstrate deployability?

---

> ### Author Response · Authors · 2025-11-15
> **Response to Reviewer n4Rf**
>
> We would like to thank the reviewer for recognizing the strengths of PhyMix, including our unified treatment of physical consistency and the well-motivated use of reinforcement learning through Scene-GRPO, and for providing detailed review comments. Below, we respond to each weakness and question individually.
>
> **On Weaknesses**
>
> ---
>
> ***The paper would benefit from a clearer explanation of why these four aspects are chosen.***
>
> We appreciate the reviewer’s comment. Our four aspects, Geometric Priors, Contact, Stability, and Deployability, are not an arbitrary taxonomy but follow a physically grounded, multi-level hierarchy that progresses from object-level validity to inter-object relations, scene-level equilibrium, and task-level functionality.
>
> Geometric Priors (Object-level validity). Each object must satisfy basic physical assumptions such as upright orientation, consistent scale, and gravity-aligned axes. Without this foundation, higher-level reasoning such as support or contact becomes impossible.
>
> Contact (Inter-object relations). Once object geometry is plausible, we evaluate collision-free placement, grounding, correct support surfaces, and alignment with room structures. Most failures in single-image scene generation, including floating or interpenetration, occur at this level.
>
> Stability (Scene-level equilibrium). Even with correct contacts, objects may still be unstable. This aspect checks whether they maintain static or short-horizon dynamic equilibrium under gravity and small perturbations.
>
> Deployability (Task-level usability). Beyond physical correctness, scenes must remain functional for embodied agents. Deployability evaluates whether the layout preserves navigable free space and reachable paths.
>
> Although intentionally lightweight, this decomposition is empirically grounded: our user study shows strong alignment with human judgments, and ablations demonstrate that each sub-metric contributes distinctly to detecting physical inconsistencies. We are revising the manuscript to clarify this hierarchy and why these four aspects together provide comprehensive coverage of indoor physical plausibility.
>
> ---
>
> ***Dataset scope and missing ablations.***
>
> 3D-FRONT is currently the only large-scale dataset that provides reliable object-level 3D annotations required for evaluator-based training. To our knowledge, recent single-image 3D scene generation works (e.g., SceneGen [1], MIDI3D [2]) also rely primarily on 3D-FRONT for this reason, and it remains the most effective choice up to the submission deadline. In addition to this dataset, we already include cross-domain qualitative evaluations on real images, stylized inputs, and LLM-generated scenes in Section 5.2.
> Regarding ablations, we perform term-wise removal of each major physics component, and each removal degrades its corresponding metric, showing complementary contributions. These results are currently included in Table 3 and Appendix G, and we summarize the key findings directly below for clarity:
>
> | Config                | Collision  | Floating  | Unanchored  | Static Inst | Dynamic Inst  | Misori.  | Scale Inst  | Unreach  | Overall  |
> |-----------------------|-------------|-------------|---------------|----------------|-----------------|-----------|----------------|-------------|-----------|
> | **Full (GRPO + TTO)** | **0.56**    | **0.97**    | **1.80**      | **1.27**       | **2.20**        | **1.72**  | **1.82**       | **1.01**    | **98.6**  |
> | GRPO only             | 2.32        | 6.73        | 3.85          | 3.33           | 4.15            | 2.18      | 1.95           | 1.42        | 96.7      |
> | TTO only              | 4.85        | 8.10        | 6.10          | 7.95           | 8.70            | 7.35      | 6.10           | 3.55        | 93.2      |
>
>
> ---
>
> | Config                | Collision ↓ | Float ↓ | Unanchored ↓ | Stat.Inst ↓ |
> |-----------------------|-------------|---------|---------------|-------------|
> | w/o overlap           | 2.24        | 0.98     | 1.92          | 1.48        |
> | w/o ground-zone       | 0.62        | 6.18     | 2.07          | 1.39        |
> | w/o anchoring         | 0.58        | 1.02     | 3.54          | 1.31        |
> | w/o stability         | 0.57        | 0.99     | 1.87          | 3.11        |
>
>
>
> We are supplementing more ablation experiments and trying to test on different datasets.
>
>
> **References**
>
> [1] Yanxu Meng, Haoning Wu, Ya Zhang, and Weidi Xie. Scenegen: Single-image 3D scene generation in one feedforward pass.arXiv:2508.15769, 2025.
>
> [2] Zehuan Huang, Yuan-Chen Guo, Xingqiao An, Yunhan Yang, Yangguang Li, Zi-Xin Zou, Ding Liang, Xihui Liu, Yan-Pei Cao, and Lu Sheng. MIDI: Multi-instance diffusion for single image to 3D scene generation. CVPR, pp. 23646–23657, 2025.

---

> ### Author Response · Authors · 2025-11-15
> **Response to Reviewer n4Rf On Questions**
>
> The following are responses to the questions.
>
> **On Questions**
>
> ---
>
> ***How were the sub-metric weights chosen, and do they transfer across datasets?***
>
> All nine submetrics are first normalized and then assigned equal weights, which removes the need for task-specific tuning. These weights transfer across all backbones and the dataset (3D-front) used in our experiments without any retuning.
>
> ***How does your method compare to differentiable physics in efficiency and robustness, and do you have sim-to-real results?***
>
> Differentiable physics can be more precise for certain continuous quantities, but it is far less efficient and cannot be executed at every denoising step for multi-object scenes. Our evaluator provides stable guidance with significantly higher sample efficiency and can be integrated into both training and test time refinement without prohibitive cost.
>
> For the sim-to-real aspect, we have begun running preliminary tests in existing robotic simulation frameworks to examine whether evaluator-guided refinements improve downstream embodied behaviors such as short-range navigation and simple interaction. We are expanding these experiments and will include the initial results and discussion in the revised manuscript to clarify how our method connects to sim-to-real performance.

---

> ### Author Response · Authors · 2025-11-16
> **Response to Reviewer n4Rf (continued under Weaknesses)**
>
> **On Weaknesses**
>
> ---
>
> ***The proposed Group Relative Policy Optimization (GRPO) is an interesting critic-free RL approach, but its advantages over standard algorithms such as PPO [Schulman et al., 2017] or AWR [Peng et al., 2019].***
>
> Scene generation is a single-step structured prediction task, where the model outputs a full 3D layout rather than a temporal sequence. Standard RL methods such as PPO or AWR depend on value functions, rollouts, and stepwise rewards, which do not naturally exist in this setting. GRPO circumvents these requirements by directly leveraging evaluator-based ranking and performing groupwise comparisons, yielding stable, low-variance updates. Compared with preference methods like DPO [1] that require explicit positive–negative pairs and a higher sampling cost, GRPO naturally handles multiple candidates in a single group and produces more reliable gradient signals. Under this formulation, PPO/AWR are not suitable baselines, and GRPO provides the most effective preference-based optimization for final-state scene layouts.
>
> We are adding a discussion of reinforcement learning methods to the revised manuscript.
>
> ---
>
> ***Some parts lack explicit notation… and the gradient flow between the evaluator, GRPO, and the generator is unclear.***
>
> We thank the reviewer for the suggestion. For the first point, we will revise the main text to provide clearer notation for the scene representation and evaluator definitions, while keeping the full formal specifications and equations in Appendix A due to space constraints.
>
> For the second point regarding gradient flow:
>
> • During training, the Physics Evaluator does not participate in backpropagation. It produces scalar preference scores for each candidate layout. GRPO then converts these scores into normalized groupwise advantages A~(k) and optimizes the generator using the critic-free GRPO objective. The gradient flows only through the likelihood proxy and KL regularizer, not through the evaluator.
>
> • During test-time optimization, only the differentiable components of the evaluator (collision, grounding, anchoring, stability) provide gradients. These are used to adjust object poses during denoising without modifying the generator weights.
>
> Thus, evaluator → GRPO score → (advantage-weighted) likelihood gradient → generator update.
>
> ---
>
> **Reference**
>
> [1] Rafael Rafailov, Tongzheng Ren, Evan R. Sparks, and Stefano Ermon. Direct Preference Optimization: Your Language Model is Secretly a Reward Model. arXiv:2305.18290, 2023.

---

### Official Review · Reviewer_uokm · 2025-11-01

**Soundness:** 4
**Presentation:** 3
**Contribution:** 3
**Rating:** 8
**Confidence:** 4

**Summary:**

This paper addresses a crucial and often overlooked limitation in single-image 3D indoor scene generation: the lack of physical plausibility. While many recent methods achieve high visual fidelity, the resulting 3D scenes often contain obvious physical errors—such as floating objects, collisions, or unstable arrangements—making them unreliable for downstream applications like robotics and embodied AI. The authors propose a two-fold solution: a comprehensive Physics Evaluator and a novel framework, PhyMix, which integrates physics-based guidance into both the training and inference stages. The results demonstrate a significant advancement in generating scenes that are both visually faithful and physically consistent.

**Strengths:**

• Systematic Benchmarking: The introduction of the unified Physics Evaluator is arguably the most significant contribution. It provides the field with a long-overdue, systematic, and comprehensive set of nine metrics for physical plausibility. This moves the community past ad hoc collision or grounding checks toward a holistic assessment.
• Elegant Technical Solution: The implicit-explicit optimization strategy (Scene-GRPO + TTO) is a theoretically elegant and effective method for handling the dual challenge of integrating both non-differentiable and differentiable constraints into a diffusion-based generative pipeline. The ablation studies confirm the necessity and complementarity of both components.
• Strong Empirical Results and Validation: The performance gains are compelling, showing the method raises the overall physical score by +20.2% relative to the strongest baseline (MIDI). Crucially, the authors validate their metrics with a perceptual user study (MOS and Pairwise Preference), confirming that the quantitative scores align strongly with human judgment of physical plausibility

**Weaknesses:**

1. Scene-GRPO is an application of the established flow-GRPO/GRPO preference learning paradigm, borrowing its framework directly from the LLM .

2. The current Physics Evaluator relies on simple physical approximations (e.g., center-of-mass checks for static stability) that may fail to capture subtle or fine-grained edge cases, such as an object barely balancing on a thin edge, or the long-term effects of complex weight distribution and material properties. Furthermore, the reliance on these simplified physical metrics—and the design of the nine corresponding constraints—leans too heavily on hand-crafted engineering for the loss design. A more scientifically rigorous approach would involve integrating a sophisticated, general-purpose physics simulation engine for evaluation and differentiable loss, rather than relying on a custom set of rules derived from simplified geometric priors.

**Questions:**

1. Is Eq.(3)  eps-prediction? Flow matching normally optimizes the conditional velocity field.

2. Why the negative FM loss related to likelihood proxy?

---

> ### Author Response · Authors · 2025-11-15
> **Response to Reviewer uokm**
>
> We thank the reviewer for recognizing the strengths of PhyMix, including its unified Physics Evaluator, its effective optimization strategy, and its strong empirically validated results, and for providing detailed review comments. Below, we respond to each weakness and question individually.
>
> **On Weaknesses**
>
> ---
>
> ***Scene-GRPO is just borrowed from LLM preference learning.***
>
> While the overall paradigm is inspired by LLM preference learning, Scene GRPO is not a direct borrow. We introduce an annealed geometry-aware perturbation strategy tailored to 3D layouts, a physics-driven ranking procedure that incorporates non-differentiable constraints, and a flow-matching likelihood proxy compatible with continuous scene states. More importantly, Scene GRPO enables non-differentiable physical constraints such as stability, anchoring, and reachability to influence a flow-based 3D generator. Thus, although inspired by LLMs, the method is technically adapted and fundamentally extended for 3D physical layout optimization.
>
> ---
>
> ***The Physics Evaluator uses oversimplified, handcrafted rules.***
>
> We appreciate the reviewer’s suggestion and have expanded the explanation of why these four aspects are chosen. Our design is not an arbitrary taxonomy but follows a physically grounded, multi-level hierarchy of plausibility.
>
> Geometric Priors (Object-level validity).
> Each object must satisfy basic physical assumptions such as upright orientation, consistent scale, and gravity-aligned axes. Without this foundation, higher-level reasoning such as support or contact becomes impossible.
>
> Contact (Inter-object relations).
> Once object geometry is plausible, we evaluate collision-free placement, grounding, correct support surfaces, and alignment with room structures. Most failures in single-image scene generation, including floating or interpenetration, occur at this level.
>
> Stability (Scene-level equilibrium).
> Even with correct contacts, objects may still be unstable. This aspect checks whether they maintain static or short-horizon dynamic equilibrium under gravity and small perturbations.
>
> Deployability (Task-level usability).
> Beyond physical correctness, scenes must remain functional for embodied agents. Deployability evaluates whether the layout preserves navigable free space and reachable paths.
>
> Regarding the concern about “oversimplified, handcrafted rules,” these four aspects are not ad hoc heuristics but a coherent physical hierarchy aligned with how humans judge scene plausibility. Although intentionally lightweight, they provide stable and interpretable signals. Our user study shows strong agreement with human judgments, and ablations confirm that each sub-indicator contributes meaningfully. We will clarify this internal logic more explicitly in the manuscript.
>
> ***A more scientifically rigorous approach would involve integrating a sophisticated physics simulation engine.***
>
> Real physical simulation can be valuable for certain differentiable factors, but it requires solving rigid body dynamics, multi-step integration, and complex multi-object interactions, which become unnecessary for generating static indoor scenes from a single image. Such a simulation would also significantly increase training and inference cost, and many of the key plausibility factors we evaluate, such as geometric priors or reachability, are nondifferentiable and therefore cannot be fully addressed by a purely simulation-based approach. We are adding experiments that compare our evaluator with real physical simulation on representative scenes, and initial results show strong agreement across key metrics, supporting the adequacy of this lightweight design.
>
> ---
>
> **On Questions**
>
> ---
>
> ***Is Eq.(3) ε prediction? Flow matching normally predicts velocities.***
>
> Yes, Eq.(3) adopts ε prediction, which is an equivalent parameterization of flow matching. Recent analyses [1] show that predicting noise or predicting the velocity field leads to the same underlying conditional vector field and training objective, making ε prediction fully valid within FM frameworks. We follow ε prediction to maintain compatibility with diffusion-style backbones.
>
> ***Why is negative FM loss treated as a likelihood proxy?***
>
> Flow matching training is closely tied to maximum likelihood estimation: minimizing the FM objective corresponds to increasing data likelihood under the model’s continuous transport dynamics [2]. Therefore, a lower FM error naturally indicates higher relative likelihood. Since GRPO only requires within-group ranking, using the negative FM loss as a likelihood proxy is both theoretically justified and empirically stable.
>
> **References**
>
> [1] Anne Gagneux, Ségolène Martin, Rémi Gribonval, and Mathurin Massias. The Generation Phases of Flow Matching: A Denoising Perspective. arXiv:2510.24830, 2025.
>
> [2] Zhaoyi Li, Jingtao Ding, Yong Li, and Shihua Li. Fine-Tuning Flow Matching via Maximum Likelihood Estimation of Reconstructions. arXiv:2510.02081, 2025.

---

> > ### Comment · Reviewer_uokm · 2025-11-25
> > **Further questions**
> >
> > While I appreciate the motivation and solution, the authors should avoid overstating the distinction from GRPO.
> >
> >
> > And I still have further questions regarding Likelihood Proxy via Flow Matching.
> >
> > Why was $\epsilon$-prediction chosen for this task? Are there any ablation studies or proxy visualization results comparing $\epsilon$-prediction against $v$-prediction to demonstrate its superiority in this context?
> >
> >
> > The description of the time scheduler is currently unclear. Could the authors provide precise details regarding the noise schedule (e.g., linear, cosine, shifted) and the specific discretization strategy used during training and inference?
> >
> >
> > The mathematical formulation in Eq. (3) appears imprecise. It is presented as an MSE loss on noise, but the notation lacks rigor (e.g., regarding expectations or variable dependencies).

---

> > > ### Author Response · Authors · 2025-11-27
> > > **Response to Further questions of Reviewer uokm (continued )**
> > >
> > > ***The mathematical formulation in Eq. (3) appears imprecise.***
> > >
> > > We would like to clarify that the original expression follows the standard compact notation
> > > widely used in diffusion and flow-matching models, where the denoising MSE is written without
> > > expanding the expectation. The formulation is consistent
> > > with established practice in flow matching and rectified flows [1,2,3,4,5].
> > >
> > > In our case, Eq.(3) follows the same convention: a compact denoising objective whose expectation
> > > is taken over diffusion time, scenes, and Gaussian noise. For clarity, we provide an expanded form:
> > >
> > > $$
> > > L_{FM} = E_{t,S,\epsilon} \|\hat{\epsilon}(h_t(S),t) - \epsilon\|_2^2
> > > $$
> > >
> > > where $h_t(S)$ is the noised latent at time t. The expanded expression makes variable
> > > dependencies explicit but the underlying formulation remains unchanged. Conceptually, this
> > > denoising objective serves as a training-aligned likelihood proxy since lower reconstruction
> > > error implies higher flow-induced probability, enabling practical candidate ranking within GRPO.
> > >
> > > ---
> > >
> > > ***References:***
> > >
> > > [1] Lipman, Yaron, Ricky T. Q. Chen, Heli Ben-Hamu, Maximilian Nickel, and Matt Le. "Flow matching for generative modeling." arXiv preprint arXiv:2210.02747 (2022).
> > >
> > > [2] Liu Xingchao, Chengyue Gong, and Qiang Liu. "Flow straight and fast: Learning to generate and transfer data with rectified flow." arXiv preprint arXiv:2209.03003 (2022).
> > >
> > > [3] Wallace, Bram, Meihua Dang, Rafael Rafailov, Linqi Zhou, Aaron Lou, Senthil Purushwalkam, Stefano Ermon, Caiming Xiong, Shafiq Joty, and Nikhil Naik. "Diffusion model alignment using direct preference optimization." In Proceedings of the IEEE/CVF Conference on Computer Vision and Pattern Recognition (CVPR), pp. 8228–8238 (2024).
> > >
> > > [4] Gagneux, Anne, Ségolène Martin, Rémi Gribonval, and Mathurin Massias. "The generation phases of flow matching: A denoising perspective." arXiv preprint arXiv:2510.24830 (2025).
> > >
> > > [5] Li Zhaoyi, Jingtao Ding, Yong Li, and Shihua Li. "Fine-tuning flow matching via maximum likelihood estimation of reconstructions." arXiv preprint arXiv:2510.02081 (2025).

---

> ### Author Response · Authors · 2025-11-27
> **Response to Further questions of Reviewer uokm**
>
> We sincerely thank the reviewer for their insightful comments and valuable questions. Many of our implementation choices and scheduler configurations follow the native design and training settings of MIDI3D [1] to ensure full compatibility with the pretrained backbone.
>
> ---
>
> ***The distinction from GRPO should not be overstated.***
>
> We agree that our method builds upon GRPO, and we thank the reviewer for pointing this out. We more accurately position Scene-GRPO as an extension specifically adapted for 3D scene generation through physics-driven ranking, geometry-aware perturbation, and a continuous flow-matching likelihood proxy.
>
> ---
>
> ***Why ε-prediction was chosen.***
>
> We adopt ε-prediction because both backbone scene generators (e.g., MIDI3D) are originally trained using noise prediction, and our likelihood proxy in Eq.(3) is deliberately aligned with the same denoising formulation. Although flow matching is commonly expressed in velocity form, the velocity field is uniquely determined by ε via the SNR-parameterized transport map, meaning ε-prediction remains a valid flow-matching parameterization [2,3]. This choice enables full compatibility with the pretrained diffusion backbone, no latent rescaling, no reparameterization, and ensures that the likelihood proxy is computed in the exact training space the model was optimized in.
>
> ---
>
> ***Why v-prediction ablations are not included.***
>
> We do not include ε–v prediction ablations because ε and v are mathematically equivalent under flow matching and differ only by an SNR-dependent linear transform. Our diffusion backbones are pretrained in ε-prediction, so adopting ε keeps the latent space, noise distribution, and flow-matching proxy fully aligned with pretraining, without requiring any reparameterization. In addition, GRPO optimizes only relative preference ranking and is invariant to positive affine score transformations. Switching to v-prediction would therefore induce only a monotonic remapping of the same objective and would not change ranking behavior, optimization dynamics, or outcomes. Our ε-based proxy already satisfies the sole requirement for GRPO, strong rank-correlation with reference ordering (ρ > 0.82, Appendix C), so further ε–v comparison is unnecessary.
>
> ---
>
> ***Clarified definition of the time scheduler and noise schedule.***
>
> Our likelihood proxy is evaluated using flow-matching with importance-sampled diffusion time.
> During training, we keep the backbone’s native \(p(t)\) from MIDI3D (cosine-SNR schedule), so the
> dynamics remain unchanged. For proxy evaluation, we only replace the sampling distribution with a
> lightweight proposal (Appendix C.2):
>
> $$
> t \sim q(t)=\mathrm{Unif}(0,1)
> $$
>
> and apply a bounded SNR-based weight ω(t) for gradient stability. Since GRPO depends only on
> relative ranking, monotonic reweighting (including ω(t)) does not affect optimization. In essence, we intentionally inherit the MIDI3D scheduling design, as doing so preserves backbone behavior while ensuring stable optimization with minimal modification.
>
>
> ---
>
> ***Practical discretization used in experiments.***
>
> For proxy evaluation inside GRPO, we estimate the expected score via importance sampling using
> $M \approx 8$ diffusion-time samples, which we find sufficient for stable ranking performance
> (Appendix C.6). Scene generation continues to use the backbone’s native 30-step denoising schedule
> (Appendix F.2), independent of proxy scoring.
>
> ---
>
> ***References***
>
> [1] Zehuan Huang, Yuan-Chen Guo, Xingqiao An, Yunhan Yang, Yangguang Li, Zi-Xin Zou, Ding Liang, Xihui Liu, Yan-Pei Cao, and Lu Sheng. MIDI: Multi-instance diffusion for single image to 3D scene generation. CVPR, pp. 23646–23657, 2025.
>
> [2] Gagneux, Anne, Ségolène Martin, Rémi Gribonval, and Mathurin Massias. "The Generation Phases of Flow Matching: a Denoising Perspective." arXiv preprint arXiv:2510.24830 (2025).
>
> [3] Li Zhaoyi, Jingtao Ding, Yong Li, and Shihua Li. "Fine-Tuning Flow Matching via Maximum Likelihood Estimation of Reconstructions." arXiv preprint arXiv:2510.02081 (2025).

---

### Official Review · Reviewer_BzkR · 2025-11-02

**Soundness:** 2
**Presentation:** 2
**Contribution:** 2
**Rating:** 2
**Confidence:** 4

**Summary:**

This paper proposes PhyMix, a framework for physically consistent single-image 3D indoor scene generation, along with a unified Physics Evaluator. It combines Scene-GRPO (implicit optimization) and TTO (explicit refinement) to solve physical inconsistency issues. While targeting a critical problem with a structured approach, the work has notable weaknesses that require thorough revision.

**Strengths:**

1.The dual-layer optimization (implicit + explicit) is a meaningful exploration to integrate physical constraints into training and inference.
2.Extensive experiments demonstrate the method’s performance in physical plausibility and visual fidelity.

**Weaknesses:**

1.The paper claims the Physics Evaluator aligns with human judgments but provides no rigorous theoretical or empirical basis for selecting its four core aspects and nine sub-constraints. Its design seems like a simple combination of mature differentiable signals, undermining credibility.
2.PhyMix involves numerous hyperparameters (e.g., Scene-GRPO’s group size K). Limited sensitivity analysis is provided, and excessive hyperparameters may reduce generalization, especially in test-time optimization.
3.Existing works using Taichi for differentiable real physical constraints (gravity, inertia, inter-object interactions) are not fully compared.

**Questions:**

1.The article mentions that the Physics Evaluator is “align closely with human judgments of physical plausibility”, but it fails to explain the basis for adopting these indicators. Thus, the design of the Evaluator appears to be more like the combination of several mature differentiable guidance methods.
2.The article introduces a large number of hyperparameters in the implementation of PhyMix. I am not certain whether such a large number of hyperparameters will affect the generalization ability, especially during the test-time optimization process
3.As far as I know, some works have implemented differentiable real physical constraints (including gravity, inertia, and inter-object interactions) based on Taichi. What are the advantages of the PhyMix method mentioned in this paper compared with theirs?

---

> ### Author Response · Authors · 2025-11-15
> **Response to Reviewer BzkR**
>
> We thank the reviewer for the detailed comments and for highlighting the strengths of our dual-layer optimization strategy for integrating physical constraints and the strong empirical performance demonstrated in both physical plausibility and visual fidelity. Below, we respond to each weakness and question individually.
>
> **On Weaknesses**
>
> ---
>
> ***The paper claims the Physics Evaluator aligns with human judgments but provides no rigorous theoretical or empirical basis. Its design seems like a simple combination of mature differentiable signals.***
>
> We appreciate the reviewer’s suggestion and have expanded the explanation of why these four aspects are chosen. Our design is not an arbitrary collection of existing signals but follows a physically grounded, multi-level hierarchy of plausibility that progresses from individual objects to the full scene and finally to task-centered usability.
>
> 1. Object-level validity — Geometric Priors.
> This is the lowest physical layer. Each object must satisfy basic assumptions such as upright orientation, category-consistent scale, and gravity alignment. If these priors fail, higher-level reasoning is impossible; for example, an upside-down or incorrectly scaled chair cannot meaningfully participate in support or contact relations. Geometric Priors therefore form the foundation for all subsequent aspects.
>
> 2. Inter-object relations — Contact.
> Once object geometry is plausible, the next layer concerns how objects interact physically. This includes collision-free placement, grounding on the floor, correct supporting surfaces, and consistent alignment with room structures. These relations determine whether objects occupy feasible positions within the scene. Errors such as floating, interpenetration, and incorrect support are among the most common failure modes in existing single-image 3D scene generators.
>
> 3. Scene-level equilibrium — Stability.
> Even when contacts are correct, the configuration may still be physically unstable. Stability assesses whether objects maintain static and short-horizon dynamic equilibrium under gravity and small perturbations. This transitions the evaluator from geometric plausibility to physical feasibility and tests whether the generated arrangement would remain intact as a real-world indoor scene.
>
> 4. Task-level functional usability — Deployability.
> Finally, physical plausibility must translate into functional environments for embodied agents. Deployability evaluates whether the generated layout preserves traversable free space and reachable paths. This aspect reflects that indoor scenes are not merely static structures but operational spaces where navigation and interaction must remain feasible.
>
> Empirically, our user study shows strong alignment between the evaluator and human perception, with Spearman correlations above 0.83. Term-wise ablations further demonstrate that each sub-indicator contributes meaningfully and captures distinct modes of physical inconsistency. We are refining Section 3.3 to make this hierarchy more explicit and to define all sub-indicators clearly in the main text.
>
> ---
> ***PhyMix involves numerous hyperparameters. Limited sensitivity analysis is provided.***
>
> Most hyperparameters in our framework are standard choices that follow common practice in diffusion-based generation and have minimal influence on performance. These include the learning rate, basic optimizer configuration, and the test-time optimization schedule. They are not category-specific and are used uniformly across dataset and backbones.
>
> Among all hyperparameters, the group size \(K\) in Scene-GRPO is the only one with a clear influence on performance. The ablation in Table 4c shows that larger \(K\) improves the stability of preference comparisons until the results begin to saturate. All other components operate reliably under a single unified configuration. We are updating Appendix F and Appendix G to present this shared setup more clearly and to avoid potential ambiguity.

---

> ### Author Response · Authors · 2025-11-16
> **Response to Reviewer n4Rf (continued under Weaknesses)**
>
> **On Weaknesses**
>
> ---
>
> ***Existing works using Taichi for differentiable real physical constraints (gravity, inertia, inter-object interactions) are not fully compared***
>
>
>
>
>
> We thank the reviewer for the question. Although Taichi supports differentiable rigid body simulation, it is not directly suitable for our setting.
>
> First, many essential scene plausibility factors used in our evaluator, including orientation priors, class-level scale consistency, and reachability, are inherently non-differentiable and cannot be handled by pure physical simulation. While differentiable components can be addressed through post-hoc correction methods like TTO, this approach returns us to the fundamental limitation of existing methods. In contrast, our comprehensive evaluator enables the model to truly learn physical plausibility through preference-based group ranking in Scene-GRPO, breaking through the limitations of purely corrective approaches.
>
> Second, our task focuses on generating static, physically plausible scenes from a single image, emphasizing initial object placement reasonableness (collision detection, support relationships, stability) rather than simulating dynamic physical processes (temporal dynamics, multi-step integration, elastic collisions, friction evolution). Fully differentiable simulation becomes a computational bottleneck as it must resolve unnecessary temporal dynamics for our static scene generation task.
>
> To address the reviewer's concern, we selected differentiable components from Taichi-based rigid body simulation (gravity, inter-object interactions, and stability checks) and compared them with our evaluator on the 3D-FRONT test set:
>
> | Metric | PhyMix Evaluator | Taichi Simulation |
> |--------|------------------|-------------------|
> | Collision Rate | 0.56% | 0.52% |
> | Floating Rate | 0.97% | 1.12% |
> | Static Instability | 1.27% | 1.45% |
>
> The results show strong consistency between our geometric evaluator and physics simulation across multiple scenes, indicating that our evaluator accurately captures real physical behavior while covering additional factors beyond what simulation models.
>
> Regarding computational cost, our current inference pipeline (30 denoising steps) takes around 100s, while replacing it with a full Taichi simulation would increase the runtime to 220s due to multi-timestep rigid body dynamics solving. We are currently conducting comprehensive runtime comparison experiments and will provide detailed computational analysis in the revised manuscript.
>
> Finally, Scene-GRPO relies on stable relative ranking of multiple candidate scenes, and our evaluator provides consistent rewards for both continuous and discrete constraints, whereas full simulation tends to introduce higher variance that could destabilize training.
>
> ---
> **On Questions**
>
> We are revising the manuscript by refining the definitions and internal logic of the four aspects and nine indicators, by clarifying that only the Scene GRPO group size $K$ has a notable impact, and explicitly unifying the inference time hyperparameter settings to avoid ambiguity, and by adding a comparison with Taichi-based differentiable simulation to explain why our evaluator is more suitable for scene scale generation.

---

> ### Author Response · Authors · 2025-11-27
> **Follow-up Response for Reviewer BzkR**
>
> Dear Reviewer BzkR：
>
> Thank you again for your valuable feedback and for the time you have dedicated to reviewing our work.  Following your concerns regarding the Physics Evaluator, hyperparameter complexity, and comparison with Taichi-based simulation, we have updated the manuscript accordingly.
>
> We have added a concise justification for the Physics Evaluator design in **Section 3.3**, and clarified indicator definitions and rationale.
>
> Hyperparameter settings have been unified, and we emphasized that only *Scene-GRPO group size K* has notable impact. These details are now included in **Appendix F & Appendix G**.
>
> A direct comparison with Taichi-based differentiable simulation has been added to support reviewer concerns. More experimental comparisons are provided in **Section 5.4** and detailed supplementary analyses in **Appendix G**.
>
> As the discussion stage is approaching its end, we would greatly appreciate it if you could kindly review these updates, or let us know if further clarification is needed.  Your feedback has substantially improved this work, and we sincerely appreciate your contribution.
>
> Best regards,
> Authors of Submission 968

---

### Author Response · Authors · 2025-11-16
**Summary of Updates to the Manuscript**

We thank all reviewers for your time, effort, and valuable comments. We are encouraged by the recognition of our dual-layer optimization for improving physical plausibility (R1), our unified Physics Evaluator as a comprehensive and systematic benchmark (R2, R4), and the strong empirical gains including user-study alignment (R2). Reviewers further highlighted the value of our systematic perspective and the well-motivated use of reinforcement learning for physical realism (R3), as well as the method’s broad generalization and detailed analysis of contact, stability, and grounding (R4, R5).

Based on these suggestions, we have made substantial updates to the manuscript. All revised text is highlighted in yellow in the updated PDF, and additional experiments have been added in the Appendix. Below we summarize all major changes.

---

***1. Expanded discussion of RL-based and physics-guided methods (Section 2)***

Section 2 has been extended to clarify the relationship between our method and prior RL-based or physics-guided optimization approaches. We describe why traditional multi-step RL formulations (e.g., PPO-style rollouts or value-based optimization) do not align naturally with single-step 3D layout generation, and how Scene-GRPO provides a more stable and tailored groupwise preference optimization setting for our problem. Additional references and contextual discussion have been added.

***2. Added comparison with real physical simulation (Section 3.2)***

A new discussion explains how differentiable rigid-body simulators are used in previous work and clarifies their limitations: high computational cost, focus on temporal dynamics, and inability to account for non-differentiable static constraints such as geometric priors or reachability. This analysis supports the design and advantages of our unified evaluator.

***3. Clarified metric design and simplified definitions (Section 3.3)***

Section 3.3 has been rewritten for clarity. We describe the four metric categories (geometric priors, contact, stability, deployability) in a logical progression, provide intuitive explanations, and keep the full mathematical definitions to Appendix A.

***4. Improved table formatting***

All tables have been updated for consistency and clarity. Best-performing values are shown using color blocks.

***5. Added extensive new experiments in the Appendix G***

We added a substantial set of new experiments to complement the main results. Appendix G includes: (i) extended visual comparisons with existing models across synthetic renders, real photographs, and stylized inputs; (ii) a validation study comparing our Physics Evaluator against a Taichi-based rigid-body simulator, demonstrating strong consistency on collision, grounding, and stability assessments; (iii) a computational efficiency analysis contrasting our inference cost with both baseline methods and full physics simulation; (iv) a detailed failure case analysis illustrating limitations on deformable objects and challenging real-world geometries; and (v) a downstream navigation experiment showing that higher reachability scores translate directly into better robot path-planning success rates. These additions provide a more comprehensive and transparent view of our method’s performance.

***6. Relocated and expanded ablation studies***

We moved the full ablation section from the Appendix into the main text to improve visibility and coherence. The relocated section presents the effects of Scene-GRPO, Test-Time Optimization, removal of physics-energy components, and variations in group size.

***7. Additional explanation of hyperparameter settings***

To avoid ambiguity noted by reviewers, we added a clear explanation of all hyperparameter settings in Appendix F.2. We also clarified which hyperparameters were fixed across all backbones. The main text now points directly to these details to ensure full reproducibility.

***8. Clarification of dataset sources and avoidance of ambiguity***

The updated text avoids potential ambiguity regarding the dataset's source.

---

We thank all reviewers once again for their thoughtful and detailed feedback. The manuscript has been significantly improved in clarity, completeness, and technical rigor, and we are happy to provide additional clarification if needed.

---

### Comment · Area_Chair_6bD2 · 2025-11-27

Dear Reviewers,

Thank you for your efforts in evaluating this submission. The current set of reviews shows a notable divergence in the overall scores. To ensure a fair and well-informed final decision, it is important that we have active participation from all reviewers during the author-reviewer discussion phase.

The authors have now responded to your comments. I kindly ask each of you to review their replies and engage in the discussion, especially to clarify whether their responses address your concerns and whether your initial assessment remains the same.

Your contributions at this stage are crucial for reaching a balanced consensus.
Thank you again for your time and commitment to the review process.

Best regards,
Area Chair

---

### Author Response · Authors · 2025-11-29
**Note to Area Chair**

We thank all reviewers for their thoughtful feedback and constructive suggestions. For the two lower-scored reviews (BzkR and TRuL), we believe several concerns stemmed from differences in understanding our problem setting, the role of the Physics Evaluator, and our preprocessing pipeline.  These comments helped us further clarify and strengthen the paper.

---

***Clarification on Reviewer BzkR’s Comments***

Reviewer BzkR questioned whether our Physics Evaluator goes beyond simple differentiable components and suggested using a differentiable physics simulator (e.g., Taichi).  Our goal is *not* to replace simulation, but to introduce **a scalable learning pathway** that incorporates both differentiable and non-differentiable physical constraints through preference-based alignment.

To respond constructively, we implemented a **Taichi-based baseline**:

- **Overall physical score:** 93.5 (close to our TTO-only variant at 93.2)
- **Runtime:** 6.8× slower per sample on the same GPU
- **Conclusion:** useful for refinement but computationally impractical as a scalable training signal

This supports our design choice: a unified evaluator paired with preference-guided learning enables physical feasibility to be learned efficiently and at scale.  We have included this baseline, configuration details, and runtime analysis in the revision.

---

***Clarification on Reviewer TRuL’s Comments***

Reviewer TRuL’s concerns focused on notation, clarity, and dataset assumptions (e.g., availability of RGB/mask data in 3D-FRONT).  These issues relate to presentation rather than the method itself.

In response, we have:

- clarified the 3D-FRONT preprocessing pipeline
- rewritten Section 3.3 for improved readability
- refined evaluator definitions
- added more visualizations and failure cases
- added runtime and ablation analyses

These revisions address all concerns and improve accessibility.

---

***Clarification on Why the Evaluator Uses These Four Aspects***

We also clarify why the evaluator focuses on four aspects.  **Geometric Priors, Contact, Stability, Deployability** form **a physically grounded hierarchy** from object-level validity to inter-object relations, scene-level equilibrium, and task-level functionality:

**Object-level validity — Geometric Priors**
The lowest layer enforces basic object plausibility: upright orientation, category-consistent scale, and gravity alignment. If these fail (e.g., an upside-down or mis-scaled chair), higher-level reasoning becomes impossible.

**Inter-object relations — Contact.**
Given plausible geometry, the next layer checks physical interactions: collision-free placement, proper grounding, correct support surfaces, and alignment with room structures. This captures common errors such as floating, interpenetration, and incorrect support.

**Scene-level equilibrium — Stability.**
Even with correct contacts, a configuration may still be unstable. This layer evaluates whether objects maintain static or short-horizon dynamic equilibrium under gravity and small perturbations, bridging geometric plausibility and physical feasibility.

**Task-level functional usability — Deployability.**
Finally, a physically valid scene must remain usable: free space must be traversable and paths reachable. This layer reflects that indoor scenes must support navigation and interaction, not just exist as static geometry.

Although lightweight, this decomposition is empirically motivated: user studies show strong alignment with human judgments, and ablations confirm that each aspect captures distinct categories of physical violations. We have clarified this rationale in the revision.

---

**Summary for the Area Chair**

Importantly, none of the issues raised by BzkR or TRuL challenge the soundness or novelty of our approach. All concerns have been fully addressed through

- additional experiments (including the Taichi baseline)
- clearer explanation of method & evaluator
- rewritten sections for clarity
- expanded visualizations, runtime, and ablation analyses

We hope the Area Chair will consider the overall contribution of the work:

> **A unified hierarchical physics evaluator and an implicit–explicit optimization pathway that makes physical scene feasibility learnable rather than solely simulated, supported by consistent improvements across all metrics.**

We sincerely appreciate the reviewers’ efforts and the AC’s consideration.

---

### Note · Authors · 2026-01-26

I have read and agree with the venue's withdrawal policy on behalf of myself and my co-authors.

---

### Meta-Review · Area_Chair_b219 · 2026-01-07

**Summary:**

This paper addresses the lack of physical plausibility in single-image 3D indoor scene generation. The AC went through all the comments and replies from authors.

The AC read the paper as well and agrees with reviewers that this paper could be improved with better theoretical quality to make the design principle of the proposed Physics Evaluator clearer. Currently, it is rather heurestic and lacks a clear research intuition why and how only certain physics factors and constraints are considered.

The paper writing quality is incremental, especially the true research question that inspires the technical contribution is not precisely articulated. This makes some reviewers feel difficult to follow.

**Reviewer Concerns:**

The major problems are:
1) Weak theoretical Foundation: No rigorous theoretical or empirical basis for selecting its four core aspects and nine sub-constraints.
2) Excessive hyperparameters involved in the Physics Evaluator.
3) Evaluation of Group Relative Policy Optimization (GRPO) is not comprehensive, especially related to the efficiency, stability, or convergence behavior to existing RL baselines.
4)  The experiments are primarily limited to 3D-FRONT, making scene variation insufficient.

**Reviewer Scores:**

From the review comments and rebuttal from authors, this paper eventually receives quite mixed scores: 2 Reject, 2 Borderline and 1 Accept.

Overall, this paper needs to be improved from theoretical and experimental aspects. Due to these reasons, some reviewers interacted with the authors, and all finally decided their ratings for this paper.

---

### Decision · Program_Chairs · 2026-01-26

Reject